# Neocortical localization and thalamocortical modulation of neuronal hyperexcitability contribute to Fragile X Syndrome

Ernest V. Pedapati [1,2,3 ✉], Lauren M. Schmitt[4,5], Lauren E. Ethridge[6,7], Makoto Miyakoshi [8], John A. Sweeney [3], Rui Liu [1], Elizabeth Smith[4,5], Rebecca C. Shaffer[4,5], Kelli C. Dominick[1,3], Donald L. Gilbert [2,5], Steve W. Wu[2,5], Paul S. Horn [2,5], Devin K. Binder[9], Martine Lamy [1,3], Megan Axford[1] & Craig A. Erickson[1,3]

Fragile X Syndrome (FXS) is a monogenetic form of intellectual disability and autism in which well-established knockout (KO) animal models point to neuronal hyperexcitability and abnormal gamma-frequency physiology as a basis for key disorder features. Translating these findings into patients may identify tractable treatment targets. Using source modeling of resting-state electroencephalography data, we report findings in FXS, including 1) increases in localized gamma activity, 2) pervasive changes of theta/alpha activity, indicative of disrupted thalamocortical modulation coupled with elevated gamma power, 3) stepwise moderation of low and high-frequency abnormalities based on female sex, and 4) relationship of this physiology to intellectual disability and neuropsychiatric symptoms. Our observations extend findings in $Fmr1^{-/-}$ KO mice to patients with FXS and raise a key role for disrupted thalamocortical modulation in local hyperexcitability. This systems-level mechanism has received limited preclinical attention but has implications for understanding fundamental disease mechanisms.

[1] Division of Child and Adolescent Psychiatry, Cincinnati Children's Hospital Medical Center, Cincinnati, OH, USA. [2] Division of Neurology, Cincinnati Children's Hospital Medical Center, Cincinnati, OH, USA. [3] Department of Psychiatry, University of Cincinnati College of Medicine, Cincinnati, OH, USA. [4] Division of Developmental and Behavioral Pediatrics, Cincinnati Children's Hospital Medical Center, Cincinnati, OH, USA. [5] Department of Pediatrics, University of Cincinnati College of Medicine, Cincinnati, OH, USA. [6] Department of Pediatrics, Section on Developmental and Behavioral Pediatrics, University of Oklahoma Health Sciences Center, Oklahoma City, OK, USA. [7] Department of Psychology, University of Oklahoma, Norman, OK, USA. [8] Swartz Center for Computational Neuroscience, Institute for Neural Computation, University of California San Diego, La Jolla, CA, USA. [9] Division of Biomedical Sciences, School of Medicine, University of California, Riverside, CA, USA. ✉email: ernest.pedapati@cchmc.org

Loss of the fragile X mental retardation protein (FMRP), mainly caused by the silencing of the fragile X mental retardation 1 (Fmr1) gene, leads to Fragile X Syndrome (FXS)[1]. FXS is an X-linked disorder that displays variable penetrance and variable expression with affected individuals characterized by a high prevalence of intellectual disability, anxiety disorders, communication impairments, sensory hypersensitivities, and autism. FMRP is a polyribosome-associated RNA binding protein that regulates the levels of many pre- and post-synaptic proteins[2]. FMRP indirectly regulates proteins that maintain neuronal excitability and directly interact with membrane-bound ion channels[3]. The loss of FMRP has been associated with neuronal hyperexcitability. The $Fmr1^{-/-}$ knockout (KO) mouse, for example, is susceptible to audiogenic seizures, displays increased spontaneous network spiking, and demonstrates elevation in resting-state electroencephalography (EEG) gamma power (>30 Hz)[4–7]. As the $Fmr1^{-/-}$ KO is central to preclinical development, it is essential to understand parallels in humans with the mouse model, but there is also an urgent need to extend our understanding of system-level dynamics in humans.

EEG is a highly feasible method to postulate whole-brain hypotheses in populations such as FXS, where invasive recordings are not available, and other neuroimaging may pose selection bias towards less impaired participants[8]. Previous reports of EEG activity in FXS have consistently demonstrated abnormalities in the alpha, theta, and gamma band. Changes in low-frequency power have been reported from the earliest clinical EEG studies of FXS, including elevated theta power and reduced alpha power[9–12]. Additionally, as in the $Fmr1^{-/-}$ KO, humans with FXS manifest increased gamma power[12–14]. Such low and high-frequency changes may also be linked, as individuals with FXS display an increase in theta-gamma cross-frequency power-power coupling (CFC) during resting state EEG[12]. Despite these intriguing findings, the confirmation of system-level hypotheses and clinical correlations largely have been limited by small samples ($n < 25$), underrepresentation of females, failure to ascertain mosaic status in males, and lack of source and functional network modeling.

Changes in thalamocortical activity are a system-level hypothesis that could explain these EEG findings and, thus far, underexplored in FXS and preclinical models. Thalamocortical dysrhythmia (TCD) is an electrophysiological model in which intrinsic changes in the thalamocortical system form the basis of disease-related pathophysiology. This model has been adopted to characterize dysregulation of cortical excitability in a host of neuropsychiatric conditions (i.e., epilepsy, Parkinson's disease, tinnitus, depression, and neuropathic pain)[15–17]. Magnetoencephalography and EEG signatures consistent with TCD include reduced alpha power, increased theta power, increased gamma power, and predominance of theta gamma CFC[17,18]. TCD related EEG alterations have been associated with clinical features in these disorders, and most recently in schizophrenia[19]. Though the underlying physiology of TCD is an active area of study, several reports have associated abnormal low-frequency cortical oscillations (and associated gamma power increases) with alterations in thalamic tonic or burst firing patterns[15–17,20]. Indeed, TCD physiology has been observed following thalamic deafferentation[20,21], excessive thalamic inhibition[22], and T-type calcium channels[23] and excitatory ligand-gated channels[24] pharmaceutical blockade. The latter has been directly observed following blockade of N-methyl-D-aspartate receptors (NMDAR) in reticular thalamus in brain slices[24], may have particular relevance in FXS, which is associated with NMDAR hypofunction[25] and dysfunction of Gamma-aminobutyric acid (GABA) producing neurons[26,27]. In FXS, thalamic abnormalities have been previously reported, including lower fractional anisotropy between thalamus and neocortex[28], alterations in T-type

calcium channels[29], reduced thalamic gray matter density[30], reduced thalamic GABA-A receptor density[31]. The implications of many of these structural and physiological changes are not well understood but identifying electrophysiological evidence of TCD in FXS could provide a larger context from which to interpret and investigate these findings.

Herein, we provide evidence of TCD in FXS. Compared to previous reports[14], we have substantially increased the sample size, conducted source localization, and modeled cortical regions and functional networks. We expected to confirm changes in theta, alpha, and gamma activity signatures consistent with TCD. Finally, we expected to find evidence that TCD-related alterations would demonstrate clinical correlation with core disorder features, including cognition and neuropsychiatric symptoms. These findings provide parallels to the $Fmr1^{-/-}$ KO and implicate subcortical contributions to the pathophysiology of FXS, which has thus far been under-reported in the literature.

## Results

**Participants and analysis overview**. We compared eighty seconds of artifact-free resting-state EEG data (Fig. 1) between 70 individuals with a genetic diagnosis of FXS (without seizures or on antiepileptics) and 71 age- and sex-matched typically developing control participants (Table 1). Other clinical phenotypic differences between groups were estimated by neuropsychiatric measures (Fig. 2a and Supplementary Table 1). Raw data were handled blindly, and there were no differences in preprocessing characteristics between groups (Supplementary Table 2). We divided spectral power into seven frequency bands: delta (2–3.5 Hz), theta (3.5–7.5 Hz), alpha1 (8–10 Hz), alpha2 (10–12.5 Hz), beta (13–30 Hz), and gamma1 (30–55 Hz), and gamma2 (65–90 Hz). To optimize the detection of neurogenic activity from the gamma band, we followed best practices to address myogenic contamination[32]. Unless otherwise specified, independent variables that were analyzed by linear mixed effect model included group (FXS or control), sex (male or female), and frequency band (delta, theta, alpha1, alpha2, beta, gamma1, and gamma2) as fixed effects and subject as the random effect. We have organized the results to report three primary analyses: 1) changes in spectral power, 2) evaluation of peak alpha frequency (PAF), and 3) alterations of CFC. We conclude with a summary of clinical correlations across analyses.

### Evidence of reduced alpha, increased theta, and gamma power in FXS from scalp EEG

*Scalp EEG Relative Power*. Before source modeling, we considered the temporal and spatial properties of the scalp EEG recording by examining the topographical distribution and scalp-averaged spectrograms of relative power across groups. Relative power (proportion of band power to total power) is generally reported to reduce inter-subject variation and facilitate group comparisons[33] in human studies. Scalp spectrogram and PAF: As in other examples of TCD, visual inspection revealed a global leftward shift in PAF towards the theta band in males and females with FXS (Fig. 2b). The model confirmed that PAF was significantly reduced in FXS with a significant group x sex interaction effect ($F_{1,137} = 5.597$; $p = 0.02$), but no effect of electrode region. Estimates of PAF (mean Hz ± std. error) in male participants with FXS (7.8 ± 0.17) were lower than either control males (9.2 ± 0.17) or females with FXS (8.8 ± 0.19). Scalp topography: We first examined power split by frequency bands between FXS and control groups. We found that in FXS, widespread alpha power decreases, theta power increases, and clusters of increased gamma power (Supplementary Fig. 1a).

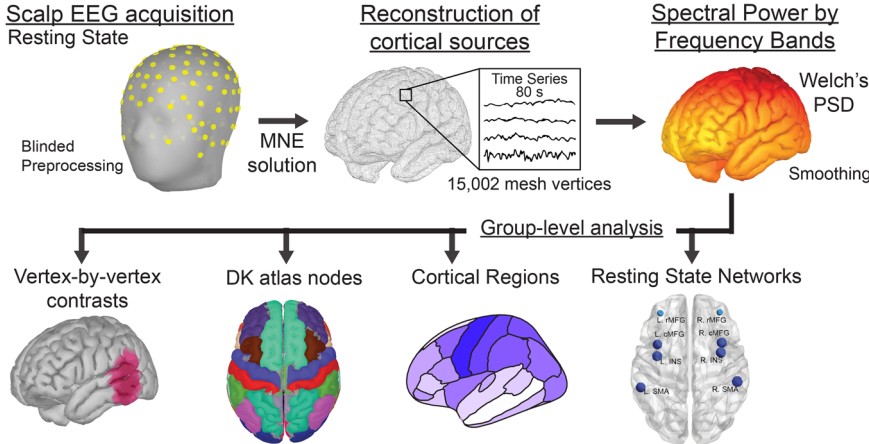

**Fig. 1 Overview of Methodology.** Principal steps of investigation: Following blinded preprocessing, artifact-free EEG data and a cortical lead field matrix were used to construct a weighted minimum norm estimate (MNE) source model to estimate current source density and spectral power by frequency band at each vertex of a triangular cortical mesh. Based on hypotheses, subsequent analyses were performed at four different levels: raw vertices, parcellation of vertex points into Desikan–Killiany atlas-based nodes, cortical regions (contiguous anatomical grouping of nodes), and/or resting-state networks (RSN) (functional non-continuous grouping of nodes).

**Table 1 Demographic and clinical features of the EEG dataset.**

| Measure | FXS (n = 70) | Controls (n = 71) | t | p | adj. p. |
|---|---|---|---|---|---|
| Age (Years) | 20.5 ± 10.0 | 22.2 ± 10.7 | <0.001 | 0.35 | 0.70 |
| Sex | F: 32 M: 38 | F: 30 M: 41 | 0.17 | 0.72 | 0.72 |
| FSIQ | 49.5 ± 30.2 | 103.2 ± 9.2 | <0.001 | <0.001 | <0.001 |
| NVIQ | 40 ± 35.7 | 103.4 ± 10.7 | −12.56 | <0.001 | <0.001 |
| VIQ | 58.9 ± 28.7 | 103.1 ± 12.8 | −10.45 | <0.001 | <0.001 |
| SCQ | 13.6 ± 7.7 | 2.1 ± 2.2 | 9.94 | <0.001 | <0.001 |
| WJ-III | 67.8 ± 15.7 | 94.1 ± 12.1 | <0.001 | <0.001 | <0.001 |
| ADAMS-Anxiety | 7.2 ± 5.0 | 2.1 ± 2.6 | 6.37 | <0.001 | <0.001 |
| ADAMS-OCD | 2.3 ± 2.5 | 0.4 ± 1.1 | 5.12 | <0.001 | <0.001 |

Showing mean (±standard deviation) group t-tests by group.
*FXS* Fragile X Syndrome, *F* female, *M* male, *FSIQ* Full Scale IQ, *NVIQ* Non-verbal intelligence scale, *VIQ* verbal intelligence scale, *SCQ* Social Communication Questionnaire, *WJ-3* Woodcock III Tests of Cognitive Abilities, *ADAMS* Anxiety, Depression, and Mood Scale, *t* t-statistic following independent Student t tests, *p* unadjusted significance, *adj. p* p value following Bonferroni correction.

*Scalp EEG absolute power.* To ascertain if these changes in relative alpha and theta power were dependent on the proportion of gamma power, we next examined absolute power in which each frequency band is considered independently. Scalp spectrogram and peak alpha frequency (PAF): The absolute power spectrogram displayed an increase in FXS participants across most frequencies (Fig. 2c). However, a significant decrease in absolute alpha2 power is visible in males with FXS. Thus, participants with FXS (exemplified in males) show similar or lower levels of alpha2 power irrespective of activity in other frequency bands. Scalp spectral power topography: Group comparison of scalp topography demonstrated elevation of absolute power, except within the alpha range where FXS and controls were generally statistically similar, but there were patches of increased alpha1 and decreased alpha2 in FXS (Supplementary Fig. 1b). The results of these analyses 1) confirm that relative power differences (especially within the alpha band) are not dependent on the proportion of gamma power and 2) the need for spectral normalization to account for large baseline differences in absolute power at the

group level. Absolute power is more influenced by head geometry[34] and skull thickness, which vary considerably across participants[33], and such factors are present in FXS[35,36]. Thus, we primarily performed relative power normalization in subsequent sections to mitigate subject- and group-specific biases, evident in absolute power analyses.

**Use of source localization to resolve scalp-level findings.** Though evidence from scalp EEG findings suggests cortical hyperexcitability, concluding the spatial distribution of these changes is limited as electrode activity represents a volume-conducted instantaneous linear mixture of underlying brain sources[37,38]. We employed source modeling to overcome the limitations of scalp EEG analysis and localize significant group differences by frequency band[39,40]. As predicted by the TCD model, we hypothesized that changes in low-frequency bands would be global, but increased gamma activity would be more localized. A depth-weighted minimum norm estimate model based on the cortical envelope of the Montreal Neurological Institute (MNI) ICBM152 common brain template was used to perform source localization[41]. The result was a triangular mesh of 15,002 vertices, with each vertex representing a time series of current source density. Vertices were parcellated into 68 cortical nodes according to the Desikan–Killiany (DK) atlas[42] for the remainder of the analyses. Two methods were used to group nodes for subsequent analysis: 1) regional (region) in which adjacent nodes were grouped to represent segments of cortical lobes, and 2) resting-state networks (RSNs) in which nodes were grouped by previously characterized highly structured EEG functional dynamics[43]. A visual atlas of the nodes is presented in Supplementary Fig. 3. Regions and RSNs are tabulated in Supplementary Table 15. Thus, analyses were performed at either the vertex, node, region, or network level based on hypotheses.

**Source localization reveals a global decrease in alpha activity and localized changes in gamma power**
*Vertex level.* We first performed a high-resolution comparison of the relative power of the cortical envelope at the vertex level to examine overall group effects. Figure 2d, e depicts 5% false discovery rate (FDR)-corrected significance between-group contrasts (FXS-Controls). In participants with FXS, we observed a global reduction in alpha2 power and a global increase in theta power. In contrast, increases in gamma activity in FXS were primarily

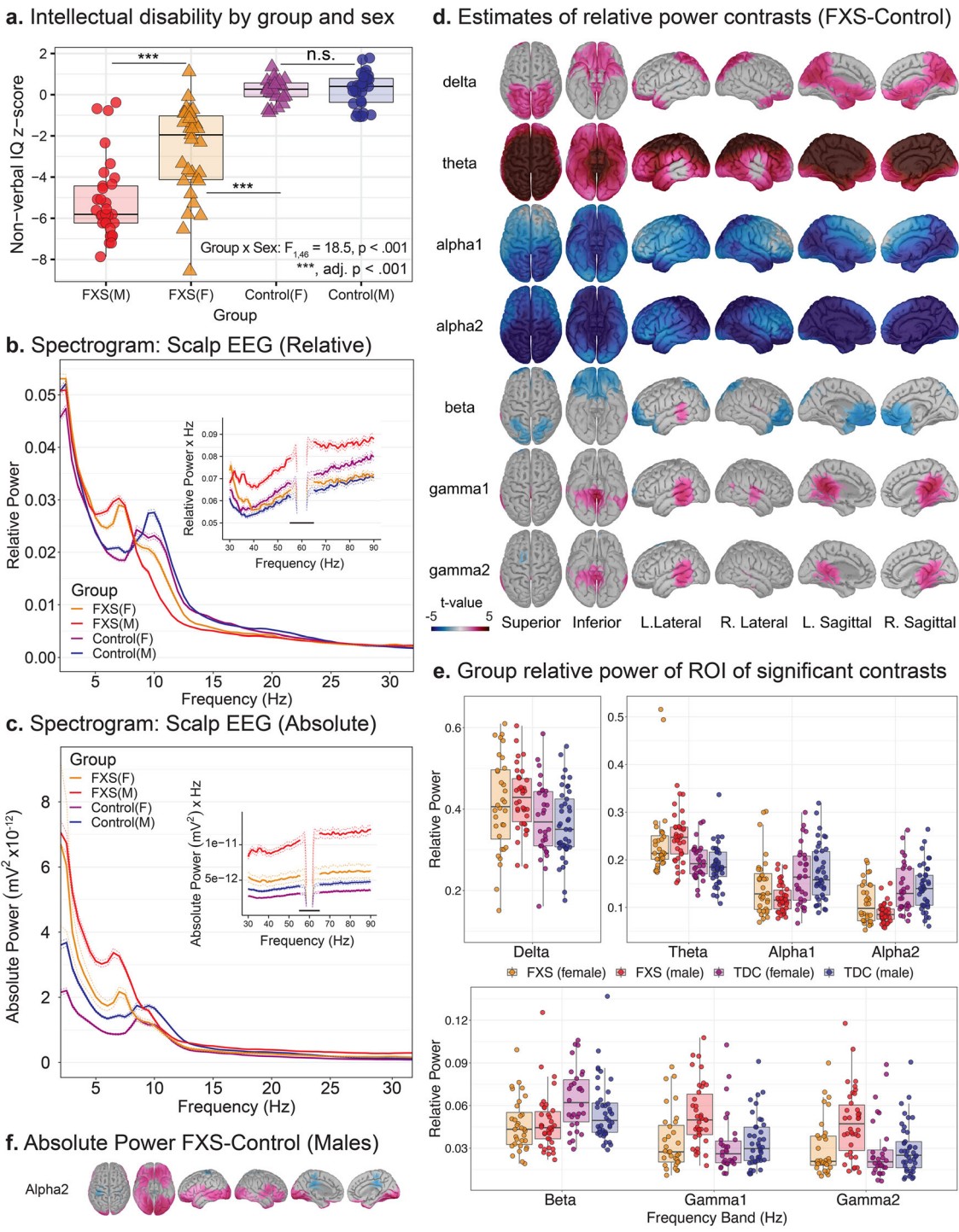

**a.** Intellectual disability by group and sex

**b.** Spectrogram: Scalp EEG (Relative)

**c.** Spectrogram: Scalp EEG (Absolute)

**d.** Estimates of relative power contrasts (FXS-Control)

**e.** Group relative power of ROI of significant contrasts

**f.** Absolute Power FXS-Control (Males)

restricted to the bilateral temporal lobes and regions of the parietal and occipital lobes. We again confirmed that the decrease in alpha2 was not due to relative power normalization by comparing source-localized absolute power between groups. Even considering that mean absolute power was increased across all frequency bands in FXS, at the group level, alpha1 and alpha2 activity were higher only in lateral regions in FXS (Supplementary Fig. 2). In a subgroup of males with FXS, absolute alpha2 power had significant clusters of increased and decreased alpha2 absolute power compared to control males (Fig. 2f). We examined additional sex effects at the region and network-level based on these significant group effects.

*Region level.* A recent study examining TCD signatures in Parkinson's disease, neuropathic pain, tinnitus, and depression found spectrally equivalent but spatially distinct forms of TCD depending on the disorder[18]. After parcellation of nodes into cortical regions, we examined the group by sex differences of spectral activity by frequency band. Based on predicted lower or absent FMRP levels and higher burden of neuropsychiatric symptoms, we expected higher deviations of spectral power in males with FXS than in controls. We found a robust interaction effect supporting our hypothesis between group, sex, frequency band, and region ($F_{78,66587} = 1.64$, $p < 0.001$). Results from pairwise comparisons of same-sex groups showed that participants

**Fig. 2 Spectral alterations are suggestive of neocortical hyperexcitability in FXS.** Scalp and source-localization of dense array electroencephalography (EEG) suggest broadband power disruptions in frequency oscillations in FXS ($n = 70$) compared to age- and sex-matched controls ($n = 71$). **a** Subject-level scatter plot of intellectual disability as estimated by non-verbal intelligence z-scores (NVIQ) by allele group. NVIQ was markedly reduced in males with FXS and to a lesser degree in females with FXS. In neurodevelopmental research, NVIQ estimates general intelligence in populations that may not have verbal communication. **b** Mean relative scalp EEG power spectrogram (thick lines) with 95% confidence intervals (dotted tracings) by group and sex and inset of depicting gamma frequencies (30–90 Hz). A marked leftward shift in dominant frequency in FXS groups and elevated gamma power. Is present. **c** Low-frequency changes in FXS groups persist in absolute power spectrograms and are not dependent on elevated gamma activity. **d** Group-level *t*-maps depicting FXS – control, vertex-by-vertex relative power differences by frequency band superimposed on brain surface models. Warmer (FXS > Control) and cooler (FXS < Control) color scale represents significant *t*-values (non-significant values as gray). **e** Boxplots displaying a median and interquartile range of relative power by frequency band (split into subplots by frequency band to optimize scale), averaged for each significant ROI from group-level t-maps from **d**. Subject level data is superimposed as a scatterplot. **f** Vertex-level comparison of source estimated absolute power in males with FXS confirms decreased alpha power is not dependent on gamma activity. For all boxplots, the center line represents the median, the lower and upper hinges correspond to the 25th and 75th percentiles), and the upper and lower whiskers extend from the hinge to the largest or smallest value, respectively, no further than 1.5 * distance between the 25th and 75th percentiles.

**Table 2 Sex and group differences in cortical regions by frequency band.**

| L/R | Region | Sex | Frequency bands | | | | | | |
|-----|--------|-----|-------|-------|--------|--------|------|--------|--------|
| | | | delta | theta | alpha1 | alpha2 | beta | gamma1 | gamma2 |
| Left | Central | F | 0.003 | 0.20 | −0.06 | −0.18 | −0.18 | −0.14 | −0.25* |
| | | M | 0.17 | 0.44*** | −0.37*** | −0.61*** | 0.02 | 0.39*** | 0.43*** |
| | Frontal | F | 0.03 | 0.21* | −0.05 | −0.17 | −0.24* | −0.29*** | −0.38*** |
| | | M | 0.06 | 0.29*** | −0.30*** | −0.48*** | −0.04 | 0.28*** | 0.29*** |
| | Limbic | F | 0.01 | 0.19 | −0.11 | −0.21* | −0.20 | −0.13 | −0.26* |
| | | M | 0.12 | 0.38*** | −0.36*** | −0.58*** | −0.06 | 0.36*** | 0.37*** |
| | Occipital | F | 0.08 | 0.15 | −0.18 | −0.26* | −0.16 | −0.02 | −0.01 |
| | | M | 0.14 | 0.28*** | −0.36*** | −0.42*** | −0.03 | 0.21* | 0.10 |
| | Parietal | F | 0.03 | 0.19 | −0.10 | −0.20 | −0.19 | −0.09 | −0.18 |
| | | M | 0.19* | 0.39*** | −0.39*** | −0.54*** | −0.04 | 0.34*** | 0.31*** |
| | Prefrontal | F | 0.05 | 0.18 | −0.12 | −0.21* | −0.31*** | −0.33*** | −0.39*** |
| | | M | 0.10 | 0.18 | −0.34*** | −0.46*** | −0.12 | 0.15 | 0.14 |
| | Temporal | F | 0.04 | 0.17* | −0.13 | −0.22** | −0.15 | −0.05 | −0.06 |
| | | M | 0.09 | 0.28*** | −0.39*** | −0.49*** | 0.05 | 0.39*** | 0.38*** |
| Right | Central | F | 0.01 | 0.18 | −0.09 | −0.18 | −0.16 | −0.09 | −0.26* |
| | | M | 0.15 | 0.42*** | −0.36*** | −0.62*** | 0.01 | 0.42*** | 0.50*** |
| | Frontal | F | 0.005 | 0.19 | −0.05 | −0.15 | −0.24* | −0.25** | −0.36*** |
| | | M | 0.04 | 0.29*** | −0.29*** | −0.50*** | −0.02 | 0.33*** | 0.39*** |
| | Limbic | F | 0.01 | 0.19 | −0.11 | −0.22* | −0.19 | −0.10 | −0.24* |
| | | M | 0.11 | 0.38*** | −0.36*** | −0.58*** | −0.06 | 0.37*** | 0.38*** |
| | Occipital | F | 0.09 | 0.14 | −0.19 | −0.25* | −0.16 | −0.03 | −0.04 |
| | | M | 0.14 | 0.28*** | −0.36*** | −0.43*** | −0.03 | 0.23* | 0.17 |
| | Parietal | F | 0.06 | 0.17 | −0.13 | −0.19 | −0.19 | −0.08 | −0.21* |
| | | M | 0.19* | 0.39*** | −0.40*** | −0.56*** | −0.05 | 0.37*** | 0.38*** |
| | Prefrontal | F | 0.06 | 0.16 | −0.14 | −0.22* | −0.32*** | −0.32*** | −0.38*** |
| | | M | 0.08 | 0.18 | −0.31*** | −0.45*** | −0.09 | 0.20* | 0.20* |
| | Temporal | F | 0.05 | 0.15 | −0.15 | −0.22** | −0.16 | −0.08 | −0.12 |
| | | M | 0.09 | 0.26*** | −0.38*** | −0.52*** | 0.02 | 0.37*** | 0.41*** |

Estimated least-squared mean differences (FXS-Control) in log relative power per cortical region by frequency band denoted with 5% false discovery rate (FDR) statistical significance. FDR (5%) adjusted significant contrasts indicated with asterisks (*$p < 0.05$; **$p < 0.001$; ***$p < 1 \times 10^{-5}$). See Supplementary Table 15 for descriptions of nodes within each region.

with FXS had increased theta and decreased alpha power across cortical regions, but this was more pronounced among males with FXS (Table 2). In addition, males with FXS, but not females with FXS, exhibited increased gamma power (Fig. 3a). Gamma power peaked across temporal regions but did not reach statistical significance in prefrontal (gamma1 and gamma2) or left occipital regions (gamma2).

*Network level.* Neuroimaging studies have identified modular brain networks related to higher-order cognitive, affective, and motor functions[44]. Thus, we extended our analysis to EEG-based RSNs, representing standing functional networks rather than contiguous anatomical regions (Supplementary Table 15). The interaction of group, sex, frequency band, and RSN was a significant linear predictor of relative log power ($F_{30,66811} = 2.21$, $p < 0.001$; Fig. 3b). Females with FXS exhibited fewer and more modest changes from control females than their male counterparts. Females with FXS, for example, had similar gamma1 and gamma2 power across auditory and visual networks and decreased gamma power across cognitive networks to control females. In contrast, males with FXS demonstrated significant elevations in gamma power across all RSNs.

**Modulation of alpha and gamma power abnormalities by sex.** We hypothesized that sex, a key determinant of phenotype in FXS[45,46], is associated with stepwise changes in EEG. We examined the above models for within-group spectral power contrasts

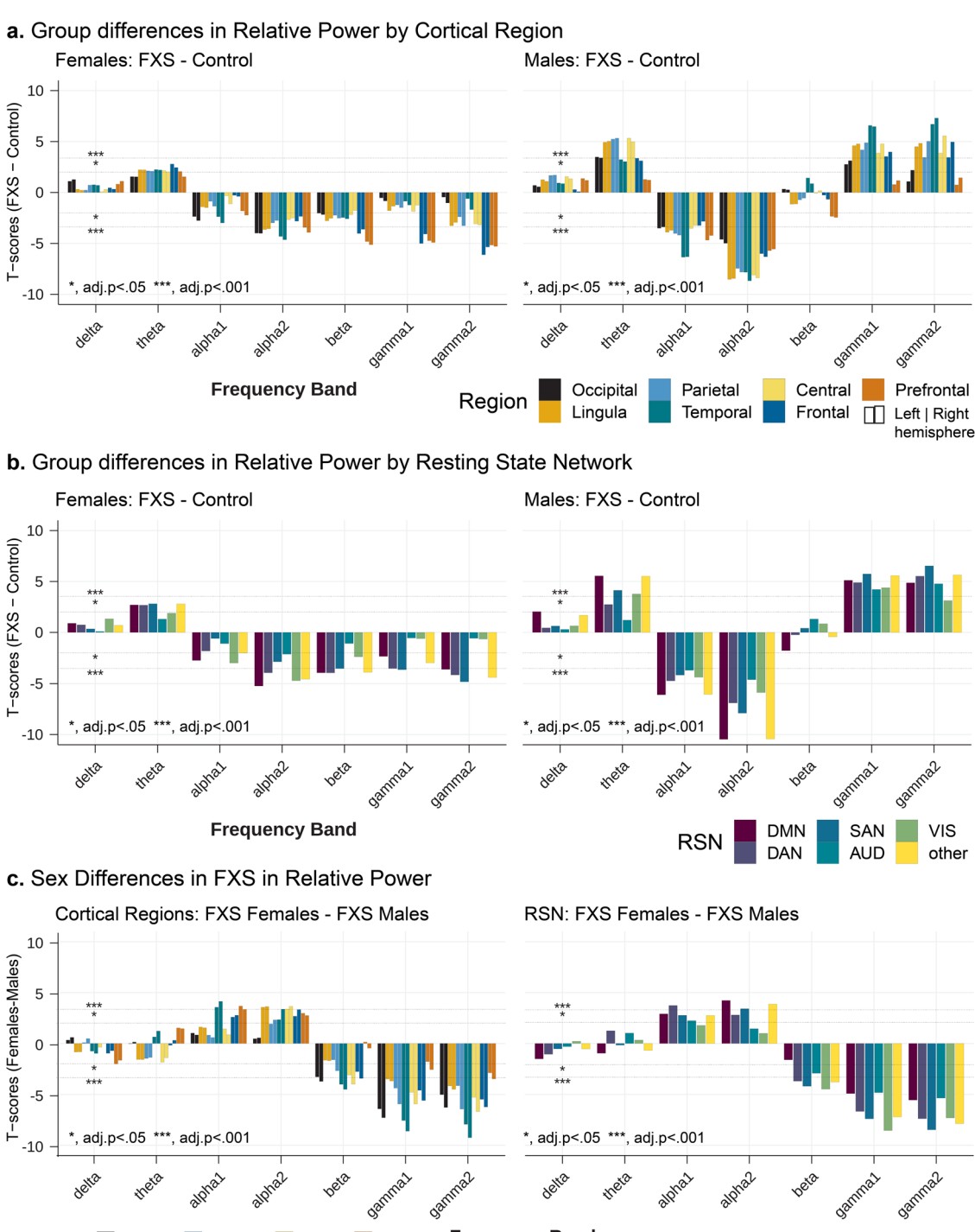

**Fig. 3 Relative power differences by cortical region or resting-state network.** Bar plots represent 5% FDR corrected pairwise contrasts of model estimates of log relative power between FXS ($n = 70$) and control ($n = 71$) participants. For region plots, left and right of individual bars correspond to the right and left hemispheres, respectively. Positive $t$-values indicate that log relative power estimates in FXS are greater than those in control. **a** Sex-matched group differences in log relative power by cortical region. Males with FXS demonstrate region-specific increases in gamma power compared to controls with sparing of prefrontal regions. **b** Sex-matched group differences in log relative power across RSNs. A significant increase in theta and gamma power, as well as a decrease in alpha power, were observed across cognitive and sensory RSNs of males with FXS. Compared to control females, females with FXS had only modest changes in RSNs including similar gamma levels in visual and auditory networks. Obligate mosaicism in *Fmr1* in females with full mutation FXS may attenuate EEG alterations. **c** We explored sex differences in relative power within the FXS group by region and RSN. Positive $t$-values indicate that log relative power estimates in males with FXS are greater than in females with FXS. There were fewer differences between males and females with FXS than in control comparisons, except for decreased alpha and that gamma activity remained elevated in most regions and all RSNs in males. Resting-state network abbreviations: DMN default mode network, DAN dorsal attention network, SAN salient affective network, VIS visual attention network, AUD auditory network.

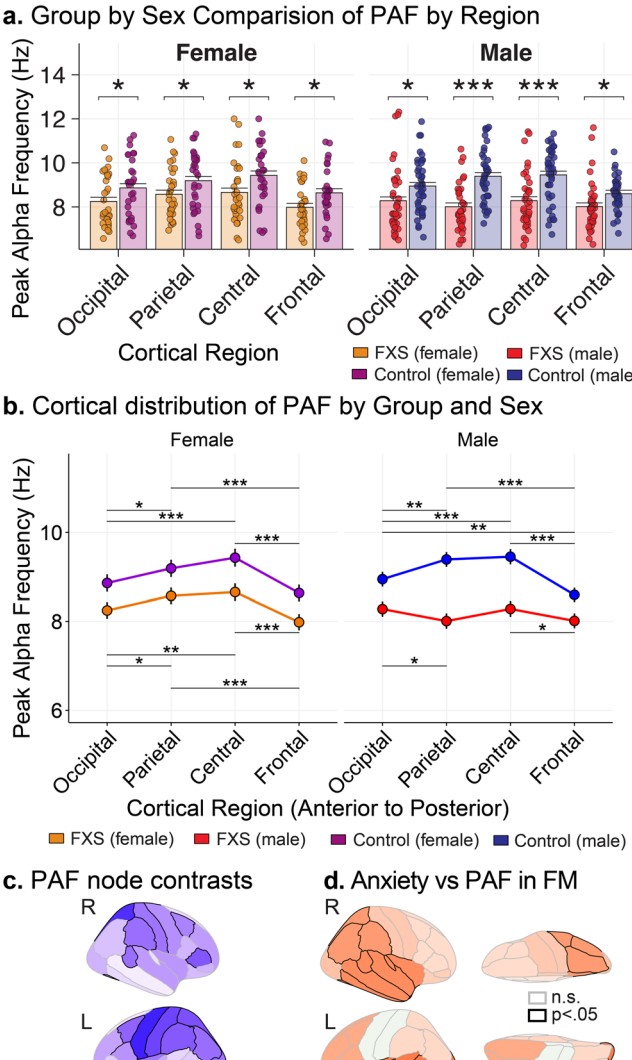

**Fig. 4 Alterations in peak alpha frequency (PAF) supportive of thalamocortical dysrhythmia in FXS patients.** The slowing of alpha oscillations have been observed across scalp potentials and intraoperative recordings and is suggestive of abnormalities in thalamocortical activity in several neuropsychiatric conditions. **a** Bar plots (model estimated mean ± standard error of least-squared mean estimates) comparing regional PAF by groups matched by sex. Scatter plot of observed subject-level PAF values are superimposed over model estimates. Participants with FXS ($n = 70$) have slower PAF across all cortical regions than controls ($n = 71$). **b** Line plots ± standard error of least-squared mean estimates comparing the within-subject distribution of PAF across posterior-anterior cortical regions. Males with FXS have a relatively even distribution of PAF across the posterior-anterior axis with no prominent central elevation as seen in matched controls. **c** Visualization of mean FXS-control differences of PAF by cortical node. Participants with FXS have broadly reduced PAF with the most prominent reductions in central and parietal nodes. See Supplementary Fig. 3 for node atlas. **d** The severity of anxiety is inversely associated with PAF. Visualization of cortex plotting age-corrected Spearman correlations in full-mutation, non-mosaic males (FM; $n = 27$; $r(25) = -0.41$ to $-0.53$, $p \leq 0.05$). FM full mutation, FXS non-mosaic males, LO left occipital. Horizontal black bars: FDR-adjusted, post-hoc testing; *adj. $p \leq 0.05$; **adj. $p \leq 0.01$; ***adj. $p \leq 1 \times 10^{-5}$.

between males and females with FXS by region and network (Fig. 3c).

*Region level.* Though theta power trended higher in males than females with FXS, no significant changes survived correction. Females with FXS generally displayed greater alpha1 and alpha2 power across regions, driven primarily by temporal, occipital, and frontal changes. Gamma1 and gamma2 power were highly elevated in males with FXS across most cortical regions (adj. $p < 0.001$), with the greatest difference in the temporal regions and modest differences in the prefrontal regions. Network Level: No significant differences in theta power were found between males and females with FXS. Interestingly, though alpha power was significantly lower in males across cognitive networks, no significant sex differences were found from alpha1 and alpha2 power across the two sensory RSNs (visual and auditory). Gamma1 and gamma2 power, however, was significantly increased in males with FXS across all RSNs ($p < 0.001$).

**Peak alpha frequency at the source level reveals reduced frequency and loss of posterior-anterior configuration**
*Region level.* Reports of TCD have observed alterations in alpha activity, including 1) leftward shift of PAF towards the theta frequency and 2) shifted spatial distribution (known as anteriorization) of slow alpha frequencies. Our goal was to determine how group, sex, and posterior-to-anterior cortical regions (occipital, parietal, central, and frontal) affect source localized PAF. A three-way interaction was present ($F_{3,4354} = 4.303$; $p = 0.005$), suggesting a linear relationship between PAF and the interaction of group, sex, and region. Frequency reduction of source PAF in FXS: To confirm the presence of reduced or slowed PAF in FXS, we examined univariate pairwise contrasts between PAF within each posterior-anterior region between sex-matched groups. Both males and females with FXS had reduced PAF compared to controls, with the largest decrease in the parietal and central regions of males with FXS (Fig. 4a). The spatial configuration of source PAF is lost in FXS: We hypothesized that PAF would be reduced and lose its characteristic asymmetrical posterior-anterior topography, as with other TCD syndromes. In male and female controls and, to a lesser extent, females with FXS, PAF was peaked over parietal and central regions (Fig. 4b). In contrast, males with FXS displayed a relatively flat profile of PAF across posterior to anterior regions. PAF in males with FXS was mildly depressed from occipital to parietal regions and not significantly different between posterior and central regions.

*Node level.* Given differences along the posterior-anterior axis, we next examined contrasts in PAF in all 68 atlas nodes to clarify the spatial distribution of PAF reduction (Supplementary Table 15). Interestingly, we found a significant interaction effect between group and node (group x node; $F_{67,9378} = 1.44$, $p = 0.01$), but no effect of sex. In approximately 51% of nodes (35/68), participants with FXS had a significantly lower PAF than controls (Supplementary Table 3). An atlas-based visualization is presented in Fig. 4c, which depicts t-scores of the FXS-control contrast. The greatest difference in PAF between FXS and controls were found in the left parietal region including the supramarginal gyrus ($F_{140} = -4.7$, 5% FDR < 0.001), inferior parietal gyrus ($F_{140} = -4.4$, 5% FDR < 0.001) and precuneus ($F_{140} = -4$, 5% FDR < 0.001).

**Theta, not alpha, power is predominantly associated with gamma power in FXS.** Power-power CFC is the association (in normalized Fisher's Z-transformed correlation coefficients) between time series of EEG power between two frequencies[47]. In TCD syndromes, theta power is more strongly correlated with

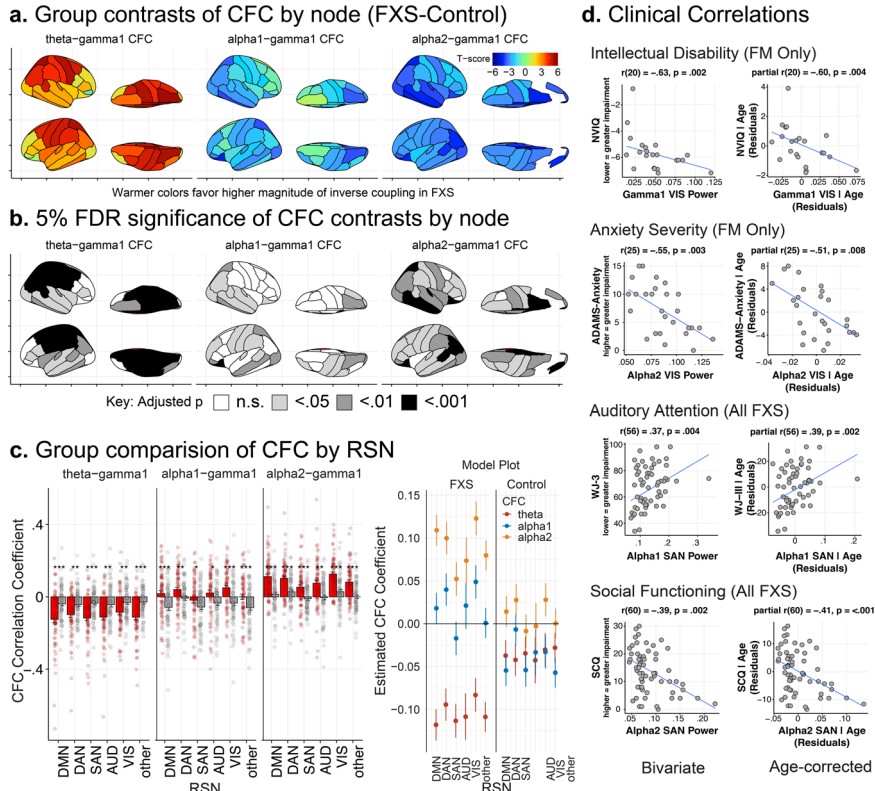

**Fig. 5 The power-power cross-frequency coupling (CFC) between theta power and gamma power in TCD syndromes is stronger than that between alpha power and gamma power. a** Brain atlas heatmaps depict group-level differences (t-values) between CFC. Since CFC coefficients can represent either a direct or inverse relationship between frequencies, the higher the *t* values, the greater the difference in the magnitude of the coupling. Participants with FXS (n = 70) showed a significantly higher magnitude of inverse theta-gamma1 CFC compared to controls (n = 71), especially over the parietal and central regions. **b** Brain atlas plots with shaded areas representing 5% FDR-corrected significance of pairwise contrasts. **c** Bar plots of mean ± standard error of least-squared mean estimates (with superimposed subject-level data) of CFC by RSN. Asterisks signify significant mean differences between FXS (red) and control (gray) groups. An accompanying mean ± standard error model plot of group by CFC type across RSN demonstrates a prominent increase in inverse theta-gamma1 CFC and an increase in direct alpha2-gamma CFC in the FXS group. **d** Exemplar clinical correlations with EEG variables across all FXS subjects and a subgroup of full mutation, non-mosaic males (FM). Scatterplots in each quadrant depict subject-level bivariate and age-corrected partial Spearman's correlations. Resting-state network abbreviations: DMN default mode network, DAN dorsal attention network, SAN salient affective network, VIS visual attention network, AUD auditory network. FDR-adjusted *p* values, *adj. *p* < 0.05; ***adj. *p*. < 1 × 10⁻⁵.

gamma power than is alpha power. These alterations have been associated with task-related disruptions in the functional activity and cognitive processing[48]. We evaluated the effects of group and sex on theta-gamma1, alpha1-gamma1, and alpha2-gamma1 CFC. As alterations of low-frequency power in FXS occur globally, we specifically examined whole-brain theta, alpha1, and alpha2 CFC with node-level and network-level gamma1 activity.

*Node CFC.* We first examined the effect of the lower frequency band (theta, alpha1, or alpha2), diagnostic group, sex, and cortical node on gamma1 CFC. Unexpectedly, sex did not have a significant main or interaction effect and was not retained in the final model. We found that the lower frequency band, group, and cortical node had a significant interaction effect on gamma1 CFC ($F_{134,28217} = 1.93$, $p < 0.0001$). Participants with FXS demonstrated an inversion of CFC relationships such that increases in theta power, not alpha, were associated with decreased gamma1 power (Fig. 5a, b). This increase was particularly manifest over cingulate, temporal, and parietal nodes (Supplementary Table 4). In contrast to controls, increased alpha2 power was directly associated with increased gamma power.

*Network CFC.* To better understand the functional impact of altered node-level CFC, we also compared CFC measured across

RSNs. We found a similar trend to our individual node analysis, with a significant interaction effect of the lower frequency band, group, and RSN ($F_{10,28589} = 3.02$, $p < 0.001$). Across all functional networks, contrasts of model estimates revealed that participants with FXS had a greater magnitude of inverse theta CFC and increased alpha2 CFC with gamma1 (Fig. 5C and Supplementary Table 5).

**Clinical correlations**. Due to the broad age range of the sample, partial Spearman's correlations were conducted to control for the effects of age on findings between EEG features and clinical measures. We first examined 5% FDR corrected correlations between (1) spectral power (Supplementary Tables 6–9), (2) PAF (Supplementary Tables 10–12), and (3) CFC (Supplementary Tables 13 and 14) values across region and RSN for all participants with FXS. FDR correction was performed on the entire correlation table for each of the three main EEG features: source estimated band power, PAF, and CFC. We next conducted an uncorrected exploratory correlation analysis on a subgroup of full-mutation, non-mosaic males (FM) to eliminate confounding with potential mosaic effects and examine findings in a more homogenous group that parallels *Fmr1*⁻ᐟ⁻ KO mouse. We have hosted an interactive statistical R web application (https://

epedapati.shinyapps.io/tcdfxs_corr/) for review and visualization. See Methods for abbreviation key for cortical regions and RSNs.

*Intellectual Quotient (IQ).* All FXS: After FDR adjustment, there was a trend-level association such that increased verbal IQ (VIQ) was associated with elevated PAF (VIQ; SAN: $r(62) = 0.39$; $p = 0.002$, adj. $p = 0.081$). FM subgroup: Reduced NVIQ was inversely associated with elevations in gamma power across multiple regions and RSNs (NVIQ; DMN, DAN, AUD, VIS, Temporal, Parietal, Occipital: $r(20) = -0.44$ to $-0.60$, $p \leq 0.05$; for exemplar Fig. 5d). CFC demonstrated significant correlations with NVIQ and VIQ. A larger relationship between increased theta and gamma1 power (theta-gamma1 CFC) was associated with increased NVIQ (SAN, DAN: $r(20) = 0.44$ to $0.49$, $p \leq 0.05$). Unexpectedly, increased association of alpha1 (alpha1-gamma CFC: DAN, AUD, VIS: $r(20) = 0.44$ to $0.53$, $p \leq 0.05$) and alpha2 (alpha2-gamma CFC: DAN: $r(20) = 0.46$, $p \leq 0.05$) with gamma1 power was associated with increased NVIQ as well. For VIQ, greater inverse alpha-gamma CFC (e.g., higher alpha power correlated with decreased gamma power) was correlated with higher verbal ability (alpha1-gamma1: LT: $r(20) = -0.56$, $p \leq 0.05$; alpha2-gamma1: LT: $r(20) = -0.56$, $p \leq 0.01$).

*Anxiety and OCD.* The Anxiety, Depression, and Mood Scale (ADAMS) anxiety and obsessive subscales assess the burden of these symptoms in individuals with intellectual disability, which are known to be prominent in FXS[49]. Higher scores indicate more severe symptoms on subscales. All FXS: Higher levels of obsessive-compulsive symptoms were associated with lower alpha1 and alpha2 power across most cortical regions and RSNs (ADAMS-OCD: $r(62) = -0.38$ to $-0.52$; adj. $p \leq 0.05$). FM subgroup: Higher anxiety levels were associated with lower alpha1 (RPF, RT; $r(25) = -0.45$ to $-0.48$; adj. $p \leq 0.05$) and alpha2 (RPF, LPF, RT, LT, RO, LO, and VIS; ADAMS-Anxiety $r(25) = -0.48$ to $-0.60$; $p \leq 0.01$ to $0.05$; for exemplar Fig. 5d) levels across several regions and within the visual network. Higher anxiety levels were associated with lower PAF across multiple regions and the DMN (RPF, LPF, RT, LT, LO, RP, and DMN: $r(25) = -0.41$ to $-0.53$; $p \leq 0.05$; Fig. 4d for correlation by region). Higher anxiety scores were associated with increased alpha1-gamma1 CFC (RO, VIS; ADAMS-Anxiety: $r(25) = 0.41–0.44$; $p \leq 0.05$). Similar directional effect was present for obsessive-compulsive symptoms with theta-gamma1, alpha1-gamma1, and alpha2-gamma CFC (RO, VIS; ADAMS-OCD: $r(25) = 0.39–0.40$; $p \leq 0.05$).

*Behavioral symptoms.* Aberrant Behavior Checklist is a widely used measure in ASD and FXS studies where higher scores indicate more severe symptoms across behavioral domains[50]. All FXS: Alpha power was inversely correlated with the abnormal speech subscale (prefrontal, SAN; $r(59) = -0.38$ to $-0.48$, adj. $p \leq 0.01$ to $0.05$), hyperactivity (AUD, DAN, PF, LT, RT; $r(59) = -0.40$ to $-0.49$, adj. $p \leq 0.05$), and stereotyped behaviors (PF, RT, LT, RO, LO, DMN, DAN, and VIS; $r(59) = -0.39$ to $-0.50$, adj. $p \leq 0.05$). FM subgroup: Elevations of gamma power were significantly associated with increased severity in behavioral domains, including abnormal speech (RPF, LPF, RT, RP, RL, LL, DMN, DAN, SAN, AUD; $r(22) = 0.36$ to $0.57$; $p \leq 0.05$), hyperactivity (LF, RP; $r(22) = 0.42$ to $0.43$, $p \leq 0.05$), irritability (RP $r(22) = 0.42$, $p \leq 0.05$), and lethargy/social withdrawal (LF; $r(22) = 0.43$, $p \leq 0.05$). Reduced alpha activity was associated with more severe symptoms including abnormal speech (alpha1: LPF, RPF; $r(22) = -42$ to $-0.49$, $p \leq 0.05$), stereotypy (alpha1: LPF, RO, RPF; $r(22) = -0.43$ to $-66$, alpha2: LPF, RPF; $r(22) = -0.48$ to $-0.55$), lethargy (alpha1: LPF, RPF; $r(22) = -0.45$ to $-0.51$, $p \leq 0.05$, alpha2: LPF, RPF; $r(22) = -0.44$ to $-0.49$, $p \leq 05$).

*Social communication.* The Social Communication Questionnaire is a brief screening instrument to evaluate social-communication skills across the lifetime where higher values indicate greater impairment[51]. All FXS: More impaired social-communication functioning was associated with decreased alpha1 (LPF, RPF, LT, RT, DMN, DAN, SAN, and VIS; $r(60) = -0.39$ to $-0.41$; adj. $p \leq 0.05$) and alpha2 power (RPF, LPF, RC, LF, RL, LL, DMN, DAN, and SAN; $r(60) = -0.37$ to $-0.52$; for exemplar Fig. 5d). FM subgroup: Greater impairments in social-communication functioning were associated with alpha1 (RPF; $r(22) = -0.43$; $p \leq 0.05$) and alpha2 (RPF; $r(22) = -0.46$; $p \leq 0.05$).

*Auditory attention.* Selective auditory attention (including speech-sound discrimination and resistance to auditory-stimulus) was measured using the Woodcock-Johnson III auditory attention task, with higher scores indicating better performance[52]. All FXS: We found that improved auditory attention performance was associated with alpha1 power (SAN; $r(56) = 0.39$, adj. $p \leq 0.05$; for example Fig. 5d). FM subgroup: Reduced auditory attention performance was associated with decreased alpha1 (LO; $r(21) = 0.52$; $p \leq 0.05$), but increased theta (LO, RO, LP, and RP; $r(21) = 0.45$ to $0.52$, $p \leq 0.05$), gamma1 (LO, RP, and VIS; $r(21) = -0.42$, $p \leq 0.05$), and gamma2 power (LO, RO, RP, and VIS; $r(21) = -0.44$, $p \leq 0.05$).

## Discussion

In the present study, we raise several findings derived from resting-state EEG to provide a unifying thalamocortical model to explain alterations of neural activity in FXS. The findings offer spatial and functional context to previously reported scalp-level EEG abnormalities in alpha/theta oscillations and elevated non-myogenic gamma activity. First, we observed clinically associated alpha and gamma activity increases, which varies by sex. Second, we report global changes in alpha and theta power, including a reduction and altered spatial configuration of PAF. Third, we demonstrate a predominance of regional and RSN-based theta-gamma CFC in FXS. These results were obtained from a well-powered sample of individuals with FXS and age- and sex-matched controls, using source modeling to identify effects with anatomical and functional ROIs. Our results highlight system-level features to enhance the development of patient-oriented biomarkers and provide key physiological insight into the neural activity of a prototypical monogenetic neurodevelopmental condition[53].

The global increase in theta power, alpha power decrease, and marked PAF reduction into the theta band implicate changes in thalamocortical circuits in FXS. In addition, the association of these changes with increases in gamma activity suggests that thalamocortical modulation may be a key mechanism underlying neocortical hyperexcitability observed in FXS[12,13,54]. Thalamocortical structures are interconnected by widespread excitatory connections, in which abnormalities have been associated with other examples of cortical hyperexcitability, including epilepsy[55]. The role of the thalamus has also continued to evolve from a relay station into a dynamic center for contextual modulation of cortical circuits[56]. Though large-scale direct measurements of thalamic contributions to neocortical excitability in FXS are unlikely, previous invasive experiments in humans, such as with stereotaxic EEG, have bolstered confidence that surface EEG oscillations are a proxy for thalamocortical activity[57]. In this sense, the observed EEG alterations, including alterations associated with TCD, may reflect the functional implications of the previously reported thalamic findings in FXS[28,30,31].

Previously it was assumed that alpha generators resided in the thalamus. However, recent research from invasive recordings in

humans has placed new emphasis on cortical generators while still confirming the central role of the thalamocortical system in driving alpha activity and PAF[57,58]. Specifically, a PAF between 10–13 Hz appears to have special relevance to functional brain physiology[59–64]. Like a radio receiver that can only tune specific electromagnetic frequencies, neurons and oscillatory networks also demonstrate a frequency preference[65,66]. The switch to a dominant peak rhythm at 4–8 Hz in FXS rather than operating peaks closer to 10–13 Hz may be insufficient to drive neural ensembles with an alpha preference[60,63,67,68]. Conversely, as theta dominance increases, alpha activity may lose its canonical role as an inhibitor of the circuits that provide optimal time for processing sensory and cognitive information[69,70].

EEG is an ideal method for studying real-time brain activity at high frequencies, but the addition of source modeling can dramatically improve spatial resolution at the cortical surface[40]. Increases in gamma power in patients with FXS were localized primarily to the temporal cortices and portions of the parietal and occipital lobes. Gamma oscillations hold a special interest in neurodevelopmental conditions because of their relation to cortical excitability[5,71], association with cognitive processes[72], and analogous measurability in animal models[6]. The role of gamma oscillations is increasingly nuanced, such that precise synchrony in gamma activity is contributory to higher-order cognition[61,73], and that a modest degree of asynchrony or noise represents physiological processes[74–76]. Nevertheless, asynchronous (usually broadband) gamma power, above what is typically expected, has been associated with disease states[72] as well as with reduced spike precision and spectral leakage of spiking activities in microcircuit preparations[77].

Gamma activity varied significantly based on sex and was strongly related to core cognitive and neuropsychiatric symptoms in FXS. As sex is highly predictive of phenotypic differences in FXS[45] and scalp-level EEG findings[14,78], we also expected sex to have a large effect on source-localized EEG activity. Unexpectedly, we found that females (obligate mosaics) generally demonstrated similar low-frequency alterations as males with FXS; however, males consistently demonstrated higher gamma power. Indeed, when compared to female controls or males with FXS, females with FXS showed either no difference or a modest reduction in gamma power. We raise at least two possibilities for these findings. First, though theta power (after correction) was similar between males and females with FXS, regional and network alpha power was higher in females. This is further strengthening the association of the presence of alpha power in regulating cortical hyperexcitability. Second, these findings suggest that FMRP expression is associated with variability in gamma power, despite not fully underlying local and system-level mechanisms leading to these changes. Within-group sex differences suggest that full-mutation females, who likely express more FMRP than their male counterparts, may moderate neocortical hyperexcitability as assessed by background gamma activity. Though in females with FXS, some liability to hyperexcitability may be present, as we found, increased gamma activity was a predictor of the severity of abnormal speech in both sexes. However, among the full-mutation, non-mosaic male subgroup increased gamma activity was predictive of worsening abnormal speech, hyperactivity, intellectual function, and auditory attention features.

Theta and alpha oscillations have been implicated in different aspects of cognitive control, for example[79]. Not surprisingly, across the FXS cohort, diminished alpha power and PAF was associated with a broad spectrum of neuropsychiatric features common in FXS, including auditory attention, hyperactivity, stereotyped behavior, obsessive-compulsive behavior, and social-communication skills. Interestingly, alpha power (or PAF) was not significantly associated with cognitive scores across FXS, despite higher PAF correlating with higher intelligence in control populations[80]. Hence, it appears that subsets of neuropsychiatric

symptoms may be associated with global changes in alpha activity. In contrast, other disorder-relevant features such as cognition may be more strongly related to gamma disturbances. Since the normalization of gamma activity is associated with increased FMRP expression, we speculate that the pattern of EEG correlations may reflect distinct mechanisms that underlie the variety of cognitive and behavioral phenotypes found in FXS. It may be possible to further explore these hypotheses in future studies which correlate specific EEG findings with domain-specific behavioral tasks.

As in other examples of TCD, we found that theta-gamma CFC, rather than alpha-gamma CFC, is predominant in FXS compared to controls[81]. We did not find any effect of sex in our final model, which likely reflects the similar low-frequency alterations in males and females with FXS, despite relative differences in gamma activity. It has been proposed that in TCD, theta modulation does not produce the same lateral inhibition of gamma activity as alpha oscillations. This leads to a net increase in asynchronous, or background, gamma activity and spurious neural activity[16]. Exploratory correlations in full mutation, non-mosaic males revealed significant associations with CFC between cognitive and neuropsychiatric features. Still, the interpretation of these findings remains complex and will require future experiments to parse. For example, a stronger inverse theta-gamma CFC relationship was associated with reduced cognitive scores, but we cannot infer either directionality or directly compare it with other frequency-specific CFC in the same subject. However, the results provide a rich starting point for developing hypotheses for future connectivity and causality analyses.

As the TCD model has not previously been explored in FXS, considering a theoretical framework from other disorders associated with TCD can provide future direction. In tinnitus, where abnormal sensory perception has been consistently linked with slow-wave oscillations, it has been hypothesized that decreased organized input to the thalamus (i.e., deafferentation or tonic inhibition) leads to excessive theta band activity. When this shift occurs, inappropriate activation of the sensory cortices (reflected by increased gamma activity) results in the perception of tinnitus. Though not directly measured in humans, neocortical changes in the $Fmr1^{-/-}$ KO mouse include intrinsic hyperexcitability of pyramidal neurons[3] and fast-spiking parvalbumin cell dysfunction[82] which is critical to sharpening synchronous neuronal activity[27,83]. Could these changes lead to a noisy cortex and disrupt organized feedback to the thalamus, and thus, thalamocortical signaling, result in alterations observed with EEG? In addition, T-type calcium channels play a central role in either tonic or burst firing of thalamic neurons[84] and T-type calcium blockade can modulate theta/alpha oscillations in the human brain[23]. The contribution of calcium channelopathies is increasingly recognized in intellectual disability syndromes[85], including known alterations in T-type calcium channels in FXS[29]. This raises speculation for new avenues of therapeutics, including targeting ion channels and non-invasive perturbation channels, to understand mechanisms that promote and maintain suboptimal brain states.

We cannot determine if the measured changes, which suggest neocortical hyperexcitability, are compensatory or causative, so back-translational approaches are necessary to uncover the underlying mechanisms of these biomarkers. For example, in some $Fmr1^{-/-}$ KO circuitry, compensatory mechanisms may partially restore global homeostasis[71]. Placed in a larger context, the circuit changes observed in FXS appear to disrupt more specialized circuits for higher-order cognition, emotional regulation, and sensory processing[86], rather than to a level representing a common cause of epilepsy (which is rare and mild in FXS). Thus, subpopulations in FXS with residual $Fmr1$ expression may vary in phenotype based on the extent of protein deficiency,

protein distribution, developmental period, circuit function, and neuronal type.

In general, absolute and relative low and high-frequency power changes were congruent, especially when examining group effects. Some changes, such as sex differences in FXS of theta power, were more prominent in absolute power. However, as seen in Fig. 4c, similar directional sex differences are present in theta power but did not reach statistical significance after multiple comparisons. Furthermore, close examination of scalp spectrograms (Fig. 2b, c) reveals within-band variability in theta and may support future analyses subdividing theta band into low and high components. Our use of relative power was to account for potential systematic bias due to head circumference and developmental stage while still acknowledging the complex between-group and within-group relationships regarding oscillatory power distribution both across frequencies and across spatial locations. Rather than assess detailed point-to-point connectivity, we used CFC to assess whole-brain averaged alpha or theta power to gamma power across individual cortical nodes to compare to reports of TCD in other disorders[12]. Within the scope of TCD, phase-amplitude (or phase-phase) coupling is not well-understood. Still, CFC is well-suited to answer hypotheses derived from our primary analyses and compare to reports of TCD in other disorders. Despite ascertaining mosaic status in males, males with mosaicism are a relatively small subset ($n = 12$) and underpowered for subgroup analysis. Additionally, males with mosaicism can vary phenotypically based on mosaic type (repeat number or methylation), further emphasizing the importance of a well-powered sample[46]. Although the effect of non-epileptic medications on the results cannot be ruled out, a medication naïve sample would preclude the inclusion of more severely affected individuals, given the high rate of medication use in FXS[87]. Previous EEG studies of medication effects in psychiatric populations[88], including our observations in FXS[78], do not suggest effects as we have observed.

In summary, the present findings demonstrate evidence of clinically relevant TCD physiology in individuals with FXS syndrome. The changes we have observed may contribute to and maintain abnormal cortical states that reduce functional brain connectivity and regional function necessary for optimal brain functions. The results suggest a central role of thalamocortical regulation of neocortical hyperexcitability in FXS and, thus, have implications for future therapeutic and physiological investigations. These systems-level findings may also guide back-translational approaches for developing and testing new treatments with electrophysiological biomarkers that can transfer directly from the mouse model to patient studies.

## Methods

**Participants.** The dataset included 145 participants drawn from a large federally funded human neurophysiology study in FXS (National Institutes of Mental Health U54 HD082008). Exclusion criteria for FXS (confirmed by Southern Blot and polymerase chain reaction) participants included a present history of unstable seizures (any treated seizure within one year) and scheduled use of benzodiazepines. Controls did not have treatment for neuropsychiatric illness as reported via clinical interview. All participants provided written informed consent (or assent as appropriate) before participation as approved by the institutional review board of Cincinnati Children's Hospital Medical Center. Following blinded preprocessing, three recordings were discarded from further analysis due to excessive line-noise artifact (1 FXS, 2 controls) and one due to insufficient data due to intolerance of the EEG procedure (1 FXS). The final dataset consisted of 70 participants with a genetic diagnosis of full mutation FXS (Mean age = 20.5, SD = 10; age range: 5.9–45.7; 32 females; 12 males with mosaicism) and 71 controls (Mean age = 22.2, SD = 10.7; age range: 5.9–48.2; 30 females). Females with full mutation FXS were included in the primary analyses, and effects were confirmed in supplementary analyses of male participants. Age effects were examined in each model for statistically significant fixed effects. Thirty-five FXS patients were on antidepressants, and 18 were receiving atypical antipsychotics. These and other concurrent medications were only permitted if the participant was on stable dosing for at least six weeks.

**Data acquisition and preprocessing.** Five minutes of continuous EEG data were collected while participants were seated comfortably while watching a silent video (standardized across participants) to facilitate cooperation as in previous studies[12]. Recordings were collected at a 1000 Hz sampling rate with an EGI NetAmp 400 with a 128-channel HydroCel electrode net (Magstim/EGI, Eugene, OR). Preprocessing: All data was blinded and coded regarding group, participant, or collection date. Data was exported in EGI raw format and imported into EEGLAB SET format in MATLAB (version 2018b, The MathWorks Inc., Natick, MA, USA). Raw EEG data were filtered using EEGLAB 14.1.2[89] with a 2 Hz high pass digital zero-phase filter and a 55 to 65-Hz notch filter (with harmonics removed up to Nyquist frequency of the original sampling rate) to remove line noise. Raw data were visually inspected by an assistant who excluded segments of data with a large amount of movement artifact and interpolated bad channels (no more than 5% per subject) using spherical spline interpolation implemented in EEGLAB 14. Data was average referenced. An artifact subspace reconstruction approach was carried out with the clean_rawdata function (with default parameters) to repair bad data segments of an artifact by applying a reconstruction mixing matrix from non-interpolated neighboring channels. The mixing matrix is computed from clean segments within the EEG data[90]. Blind source separation was performed with temporal Independent Component Analysis on each preprocessed dataset using the extended INFOMAX algorithm[91,92] with principal component analysis rank reduction (further reduced for interpolated channels). This approach was recently validated to effectively reduce myogenic contamination from approximately 25–98 Hz[93]. The resulting components were manually reviewed and categorized for eye movement/blinks, muscle movement, channel noise, or cardiac artifact based on temporospatial and spectral features and back-projected to remove artifact. The resulting non-artifactual independent components are near-independent in time course activity and resemble dipolar scalp projections, and have been proposed to represent spatially coherent local field activity within a single cortical area[94]. Data was divided into 2-second epochs and manually reviewed for noise artifacts. Summary of artifact cleaning is presented in Supplementary Table 2 and demonstrates no significant differences between preprocessing measures between groups.

**Source modeling.** Minimum norm estimation is a widely adopted solution to the inverse problem. Current estimates are calculated at every spatial location in source space to minimize the total power across the cortex[95]. Thus, minimum norm models, in contrast to dipole fitting, produce uniform maps across subjects which is well-suited for group comparisons and can provide resolution comparable to magnetoencephalography[96]. For each subject, the first 80-s of artifact-free data from each of the EGI 128-channel electrodes were co-registered with a Montreal Neurological Institute (MNI) averaged ICBM152 common brain template[97]. The degree of accuracy and precision of EEG source localization is debated, but intracortical recordings during epileptic surgery[98], surface and deep brain stimulation[99], and comparisons with functional magnetic resonance imaging (fMRI)[100] estimate focal localization at 1.5 cm for superficial neocortex. Thus, even with standard head models and spatial smoothing EEG is suitable for studying high-frequency brain activity in vivo clinical studies[101]. OpenM/EEG[102] was used to compute a 15,000 vertices lead-field mesh incorporating electrode distances. Noise covariance was set as an identity matrix as recommended for scalp resting EEG recordings[41]. Construction of an L2-normed, depth-weighted minimum norm source model to generate a current source density map (units: picoampere-meter) was performed in Brainstorm[41] and used to reconstruct time series activations at each vertex.

**Anatomical and functional parcellation.** Individual vertices from the lead-field mesh were grouped into 68 cortical nodes according to the Desikan–Killiany (DK) atlas[42]. We opted to study both the anatomical and functional configuration of the atlas nodes. The anatomical configuration was derived from the DK atlas. It consisted of categorizing the 68 notes into 14 regions: occipital (O), Limbic (L), parietal (P), temporal (T), central (C), frontal (F), and prefrontal (PF) each with a right (R) or left (L) designation. Functional source EEG resting-state networks have been derived from examining their dynamic properties and similarities to networks identified by other neuroimaging techniques (diffusion tensor imaging, fMRI, and magnetoencephalography)[43]. Following this template, we assigned 44 cortical nodes into resting-state networks including default mode network (DMN), dorsal attention network (DAN), salient affective network (SAN), auditory network (AUD), and visual network (VIS). The remaining nodes not associated with a functional network are classified as "Other" by convention.

**Spectral power.** For all analyses, we divided spectral power into 7 frequency bands: delta (2–3.5 Hz), theta (3.5–7.5 Hz), alpha1 (8–10 Hz), alpha2 (10–12.5 Hz), beta (13–30 Hz), and gamma1 (30–55 Hz), and gamma2 (65–90 Hz). Upper alpha bands are associated with more complex cognitive processing[70,103], and lower alpha bands have been primarily related to attentional processes, including alertness, expectancy, and vigilance[104].

**Scalp EEG.** Segmented data (2-s) from 108 scalp EEG channels were detrended, tapered with a Hanning window, and transformed into Fourier coefficients representing 0.5 Hz frequency steps. Fourier coefficients were squared to compute

absolute power and divided into frequency bins. Relative power was defined as the band-specific cumulative absolute power ($V^2$/Hz) divided by the total power across all defined bands and then averaged over available trials. For source data, Welch's method (50% overlap with Hanning window) was used to estimate spectral power from each vertex amplitude time series. To facilitate group comparisons, we used a circularly symmetric Gaussian smoothing kernel with a full width half maximum (FWHM) size of 3 mm[105] across all vertices. Relative power calculations were performed identically to scalp electrode power.

**Peak alpha frequency.** The average dominant frequency (i.e., alpha peak) was determined by the findpeak() function in MATLAB to identify the frequency of the maximum absolute logarithmic power between 6–14 Hz from each channel or DK node spectrogram[106].

**Clinical measures.** Stanford-Binet Intelligence Scale 5th Ed. (SBS)[107] was conducted by trained clinicians in both FXS and control participants. Due to floor effects, deviation IQ scores[108] were computed to capture variability in cognitive functioning. The primary caregivers completed assessments for FXS patients, including the Social Communication Questionnaire[109], Anxiety, Depression, and Mood Scale (ADAMS)[49], Woodcock-Johnson III Tests of Cognitive Abilities, Auditory Attention subscale[52].

**Statistics and reproducibility.** The dataset consisted of 70 individuals with a genetic diagnosis of FXS (without seizures or on antiepileptics) and 71 age- and sex-matched typically developing control participants. See Data Availability and Code Availability sections for access to publicly hosted datasets and analysis scripts. Eighty seconds of artifact-free resting-state EEG data for each participant was used for analysis.

**Power and sample size.** Differences in gamma1 power in FXS compared to controls in previous studies have effect sizes from 0.63 to 1.75, similar to effect sizes in prior studies of N1 amplitudes in FXS[12,13,78,110]. Based on these effect sizes, comparing 70 FXS patients (50% males) and 70 TD controls provides power to detect the primary EEG outcome with approximately power > 0.90 (using an omnibus F-test with an alpha of 0.05). In line with reproducible research guidelines, scripts for the generation of figures and tables are available upon request.

**Topographical electrode power comparison.** Cluster-based permutation analysis[111] was used to identify differences in relative or absolute power between FXS and controls. Overall alpha was set at 0.05/7 (adj. $p. < 0.007$) to account for multiple frequency band comparisons (effective alpha for each tail 0.025).

**Source model power comparison.** Group-level statistical ($t$-statistic) cortical maps were generated by Monte-Carlo permutation (2000) after independent two-tailed $t$-tests (alpha set at 0.025 per tail) using the 'ft_sourcestatistics' function in FieldTrip[112] and threshold at $p < 0.05$. The resulting $p$ values were globally corrected by a false discovery rate (FDR) of 5% applied over the signals and frequency band dimensions[113].

**Node, region, and RSN comparisons.** Log-transformed power differences were evaluated with generalized linear mixed effect models via NLME library in R4.1 and confirmed with GLIMMIX procedure in SAS 9.4 (SAS Institute Inc., Cary, NC, USA). Random effect was subject and independent variables varied based on model. Fixed effects included GROUP (FXS vs. control), SEX (male vs. female), or FREQUENCY BAND (delta, theta, alpha1, alpha2, beta, gamma1, and gamma2). When specified, NODE indicates the 68 cortical parcels of the DK atlas (Supplementary Fig. 3), REGION refers to the 14 node groups that represent cortical lobes, and RSN refers to the six functional grouping of nodes (DMN, VIS, DAN, SAN, AUD, Other). In REGION and RSN models, nodes were treated as replicates. To ensure optimal model fit, we examined various structures of intra-subject covariance and link functions. For each model, plots based on the studentized residuals were examined. See Supplementary Fig. 3 for a visual DK atlas key and Supplementary Table 15 for a comprehensive classification table of cortical nodes.

**Cross frequency amplitude coupling.** To examine the potential dependence between low-frequency power and high-frequency activity, we calculated power-power CFC[12]. CFC was calculated based on each low-frequency band's mean whole-brain power (theta, alpha1, and alpha2) compared to gamma1 power within each individual cortical node. The Spearman's correlation coefficient for each low frequency and gamma1 was calculated using the time series of mean relative power across 2-s epochs. Fisher's Z-transform was used to normalize group-wise comparisons.

**Correlation analysis.** As a successive step, frequency bands of significance were linearly correlated with clinical and behavioral measures. Primary Analysis: Shapiro-Wilk's normality test was performed on variables to assess suitability for either Spearman's rank-order or Pearson's correlation test. A priori hypotheses for high-frequency bands (beta, gamma1, and gamma2) and low-frequency bands (theta, alpha1, and alpha2) with clinical variables were assessed with correlation tests with p values adjusted by FDR for multiple test iterations, and partial correlations were used to adjust all correlations for age.

**Reporting summary.** Further information on research design is available in the Nature Research Reporting Summary linked to this article.

## Data availability
EEG datasets and figure data that support the findings of this study are available in Zenodo and Figshare with the identifier(s): https://doi.org/10.5281/zenodo.6385768[114] and https://doi.org/10.6084/m9.figshare.19424015.v13[115].

## Code availability
Scripts used for EEG analysis are available at http://github.com/cincibrainlab/vhtp. Time-stamped analysis code is available on Figshare[116].

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

## Acknowledgements

We thank the participants and families who participated in this study. We would also like to thank Nicole Friedman, Michael Hong, Danielle Chin, and Janna Guilfoyle, who assisted with the project. The present study was federally funded by the National Institutes of Health (NIH) Fragile X Centers (U54HD082008 and U54HD104461).

## Author contributions

Conceptualization: E.V.P., J.A.S., C.A.E, and D.K.B., Data collection: M.A., R.C.S, C.A.E, E.V.P., K.C.D, and C.A.E., Writing (original draft): E.V.P., R.L., J.A.S., L.M.S., Writing (review and editing): E.S., R.C.S, L.E., M.M., M.L, D.L.G., and S.W., Extensive Revision following peer review: E.VP., L.E. and M.M. Data curation: M.A., R.L., and E.V.P, Analysis and Statistics: R.L., L.E., P.S.H, and E.V.P., and Supervision: J.A.S., C.A.E. Competing interests: The authors have indicated no conflicts interest with the present data.

## Competing interests

The authors declare no competing interests.
