## [Peer Review File · Communications Biology]

Reviewers' comments:

Reviewer #3 (Remarks to the Author):

Summary: This manuscript presents data from a relatively large EEG dataset of individuals with FXS. The size of the data set is impressive in the context of the genetic disorder and neuroimaging, and the data set includes a large number of males and females allowing for analysis of differences between sex and mosaicism in FXS. Overall I found the data to be intriguing, however the organization and presentation of the results (in the text, not the figures) hindered my ability to easily understand and evaluate the breadth of the findings. From the perspective of the readers of the journal and the results influencing the thinking in the field, I recommend a major revision of the text to provide more context for analyses presented both in the introduction and throughout the results. I have provided more detailed comments below.

Further, I have methodological concerns that affect the major claims of the paper that I believe need to be addressed. The main concern that reoccurred throughout my review relates to the choice to use relative rather than absolute power for the analyses. Previous studies in adults and mice have consistently found striking differences in specific frequency bands (largely low and high frequencies) for FXS. Given this, relative power in one frequency band may be affected by specific increases in another frequency band. For example, for source analyses, if absolute beta and gamma power are increased in specific cortical region in FXS but not TDC, wouldn't that differentially affect the relative power of other frequency bands in those regions for FXS, but not the TDC group? If the FXS group simply had an overall shift in power evenly distributed across all frequency bands, using relative power would make more sense, but this does not appear to be the case.

Detailed comments by section:

Introduction:

1. Framework: A large portion of the results pertains to presenting evidence of thalamocortical abnormalities in FXS, however this is not set up in the introduction. I think it would be helpful for readers if more of the background and reasoning for this line of investigation were provided in the introduction. Similarly, there is not a clear description of the research questions and hypotheses being tested, but rather a statement of the findings. I would recommend providing more context for the analyses performed.

Methods:

1. Were there group differences in the number of seconds/epochs of artifact free data?
2. As discussed above, I would like to see more rationale for the use of relative power instead of absolute power. Data from human and mouse work finds that high frequency power (over 25 Hz) is elevated in FXS, and thus total power will also be elevated in FXS – but due to an disproportionate increase in the higher frequency – resulting in a lower relative power for lower frequencies. In this scenario, what is the functional relevance of relative power? What are the reasons for not using absolute power?

Results:

Overall, I found the results section to be clear at the beginning, but still dense to read, and progressively less clear (especially starting at the Cognitive Networks section). I recommend presenting the reasoning behind each analysis – what question are the authors hoping to address with the following analysis – to help better guide the reader through the results section. I recognize that many cognitive neuroscience papers simply state the results without background reasoning provided. Given the large amount of data and analyses presented, the current version lacks clarity in the reasoning behind the analyses and thus the digestibility of the findings is diminished. I also wonder whether the Cognitive Networks section could be presented in table format so that the text could focus more on what analysis was performed and why, and then highlights?

1. Figure 1C: Perhaps this should be plotted differently? The inset graph shows clear increased

power between 30-90hz for the FXS group, however this cannot be seen at all the right side of the larger plotted, presumably because of the y-axis scale that is used. Is there a reason this wasn't all plotted on a log scale?

I would like to see the absolute power plots as well. As discussed above, the increase in high frequency band power could lead to reduced relative power in lower frequencies, while absolute power is similar.

2. Figure 1E: I apologize if I missed this in the methods section – is power measured over the band the average relative power/Hz within the frequency range, or the sum of the relative power across the defined frequency range? It would be helpful to have a unit of power so it is clear what is being calculated here.

3. Is it standard practice to use relative power for cross frequency coupling? I am not an expert in CFC, however since you are already comparing power between two frequency bands within an individual, it seems that absolute power would be more accurate in this instance. By definition, relative power in the alpha band is normalized in part by power in the beta and gamma bands – and this may not occur equally between FXS and controls, given the increase in gamma in the FXS group. In addition, given the shift in peak from 10 to 7 hz, it would be more informative to compute comodulograms comparing low and high frequency ranges, so that the association can be more specifically analyzed across the range of frequencies.

4. Figure 2C – I was surprised that this analysis was done using a ratio of relative power approach. If the question is where topologically the shift to a lower alpha frequency occurs, why not look at the peak frequency across different regional nodes? Similar to my concern above – I'm not convinced that relative power is the best measure for this analysis, since a ratio between alpha power is already naturally normalized to itself.

5. Oscillatory activity in cognitive and affective network section: "Between and within group comparisons....." sentence is not complete.

6. Can the effects of sex be separated from the effects of mosaicism across both sexes? Given the large sample size, can you do this comparison using only females and mosaic males? I think it is important to know whether this is truly a sex differences versus a difference in protein expression. In Figure 3b, I would be interested in seeing the difference between mosaic males and full mutation ("obligate mosaic") females as they are more likely to have similar protein expression deficits.

Discussion

1. I found the shift in low frequency peak from 10 to 7 hz to be very intriguing, especially in the context of TCD. I would be interested in further discussion on authors thoughts on whether this is a slowed alpha or a shift from alpha to theta oscillations, and how conceptually these would relate to the TCD construct. On the individual level, is there a single peak, or do individuals have both theta and alpha peaks? I'd also be interested in a discussion of why amplitude-amplitude coupling rather than phase-amplitude coupling was analyzed. Phase-amplitude coupling between theta and gamma is well studied and as I understand, also associated with TCD. What information do each of these methods provide, and why chose one over the other? It seems that with the differences in high frequency power in the FXS group, use a phase approach may be helpful.

2. I would like to see a more thorough discussion of the sex differences observed, especially in the context of mosaicism in both male and females in the dataset. Authors discuss the effects of residual FMRP levels in females and how this may moderate cortical hyperexcitability, however wouldn't this also be true for mosaic males?

3. What hypotheses do authors have on why associations between gamma and cognitive scores were only present for full mutation males? Wouldn't one expect an even clearer dose response between gamma activity and cognition in the mosaic population if gamma were directly related to

cognitive development? Or does this suggest that having no FMRP production leads to different downstream brain differences than having low FMRP production?

Minor:

1. Page 2 "key illness feature", I found "illness" to be unusual descriptor as it generally is associated with disease rather than disorder. I might use a different word here – a key characteristic of the disorder? Or maybe hyperexcitability is central to pathophysiology of the disorder?

Reviewer #4 (Remarks to the Author):

The paper submitted by Pedapati et al. presents an EEG study in Fragile X syndrome patients, in which the authors want to examine the neocortical localization of thalamocortical modulation of neuronal hyperexcitability. A strength of the study is the large sample size in this rare disease. However, although I think that the authors apply sophisticated EEG methods, I do not see much justification for their analysis, motivation for the study and conclusions drawn. Below I outline the most problematic aspects.

1. To me in the introduction, the authors mainly motivate their study on the grounds of insufficiencies of previous studies (sample size, un-balanced gender, lack of network modelling). Especially regarding the latter is not clear why(!!) the network modelling is important for an advance in the conceptual understanding of FXS.
2. Related to the above point, the authors strongly drawn on the role of gamma oscillations in neurodevelopmental disorders. They seems to equal gamma oscillations with ability in higher order cognitive functions, which is not true. Lower-frequency oscillations (especially theta oscillation) may be of similar important, especially given their relevance in integrating processes between distant brain regions (as aspect relevant to FXS). The authors do not provide justification for their analysis of various frequency bands and the entire data analysis is very exploratory. Similarly, there is no conceptual justification for e.g. analysis of alpha-gamma coupling vs. theta-gamma coupling. To go on; what is the conceptual relevance for the various correlation analyses performed between EEG frequency band power and clinical parameters such as OCD symptoms, irritability etc.?
3. Generally, it is not clear why the different analyses presented in the results section were performed and how these conceptually relate to each other and the main study question at hand. I think that for clinical studies of this sort there should be clear conceptual hypotheses that should be focused. Especially considering 'replication crisis' a data-driven convenience approach as performed by the authors bears the risk that non-meaningful results become published. Generally, the results section is very difficult (if not impossible to understand), because the different steps undertaken in the data analysis are not justified/unclear. Just one example: "Prominent group differences (FXS-Control) in RSN activity by frequency band emerged at the node level in DMN". Which other networks? Why to the authors focus on the "node level"? Why is a frequency band specific analysis reasonable/justified? Why do the authors stress the DMN? What about the other networks/why are these less interesting to report in more detail? Similar lists of opaque analysis steps can be given for almost all sections of the results.
4. Large parts of the discussion section are too much speculative. For example the authors extensively discuss the role of GABAergic cells, but the data level analyzed allows little insights into this level of arguments. The authors do not convey a clear picture on how the different analyses performed convey to a better understanding of FXS. This is, I think, related to the exploratory nature of the study.
5. In relation to the above comment, the hypothetical model of thalamocortical dysrhythmia in Figure 4 is not sufficiently related to the data obtained. In fact, the authors provide data how different EEG spectra are modulated in FXS and what cortical structures are related to it. Their model in Figure 4, however, is cortical layer specific (not measured in the study) and draws on the role of thalamic connections to the cortical layers (also not measured in the study). Therefore, the theoretical impact the authors try to convey in their study is heavily overdrawn. Also, the model is

too simplified as it equals the output of layer 5 (not really defined what output it is) to intellectual disability, anxiety etc. This is too simplistic and, I think, misleading.

Overall, I think that the paper needs major reworking in terms of hypotheses and concepts being tested and conclusions drawn from the data analysis. I do not think that the work is publishable as it stands now.

Response to Reviewers

January 31, 2022

Reviewer #3: Summary

This manuscript presents data from a relatively large EEG dataset of individuals with FXS. The size of the data set is impressive in the context of the genetic disorder and neuroimaging, and the data set includes a large number of males and females allowing for analysis of differences between sex and mosaicism in FXS. Overall, I found the data to be intriguing, however, the organization and presentation of the results (in the text, not the figures) hindered my ability to easily understand and evaluate the breadth of the findings. From the perspective of the readers of the journal and the results influencing the thinking in the field, I recommend a major revision of the text to provide more context for analyses presented both in the introduction and throughout the results. I have provided more detailed comments below. Further, I have methodological concerns that affect the major claims of the paper that I believe need to be addressed. The main concern that reoccurred throughout my review relates to the choice to use relative rather than absolute power for the analyses. Previous studies in adults and mice have consistently found striking differences in specific frequency bands (largely low and high frequencies) for FXS. Given this, relative power in one frequency band may be affected by specific increases in another frequency band. For example, for source analyses, if absolute beta and gamma power are increased in the specific cortical region in FXS but not TDC, wouldn't that differentially affect the relative power of other frequency bands in those regions for FXS, but not the TDC group? If the FXS group simply had an overall shift in power evenly distributed across all frequency bands, using relative power would make more sense, but this does not appear to be the case.

Authors' Response A1:

We appreciate the reviewer's constructive feedback and agree that a thorough revision of the text was necessary to frame the significance of the results. We have thoroughly revised all aspects of the text, updated figures, and conducted new analyses. We have also addressed major organizational issues, including providing greater context to our analysis approach. We have addressed specific concerns below.

Reviewer #3: Framework (Introduction)

A large portion of the results pertains to presenting evidence of thalamocortical abnormalities in FXS, however, this is not set up in the introduction. I think it would be helpful for readers if more of the background and reasoning for this line of the investigation were provided in the introduction. Similarly, there is not a clear description of the research questions and hypotheses being tested, but rather a statement of the findings. I would recommend providing more context for the analyses performed.

Authors' Response A2:

We agree with the reviewer and have rewritten the introduction to better frame our results, including hypotheses regarding analysis parameters and a discussion of thalamocortical abnormalities in FXS. We have substantially revised the results adding greater context and hypotheses for each analysis to the findings.

Reviewer #3: Methods and preprocessing

1. Were there group differences in the number of seconds/epochs of artifact-free data?

Authors' Response A3:

No differences were found, and a group-level summary of preprocessing metrics has been added to the results (**see Table S1**).

Reviewer #3: Relative vs. Absolute Power

2. As discussed above, I would like to see more rationale for the use of relative power instead of absolute power. Data from human and mouse work finds that high frequency power (over 25 Hz) is elevated in FXS, and thus total power will also be elevated in FXS – but due to a disproportionate increase in the higher frequency – resulting in a lower relative power for lower frequencies. In this scenario, what is the functional relevance of relative power? What are the reasons for not using absolute power?

Authors' Response A4:

The reviewer raises an important concern – since gamma power is known to be elevated in FX-species, would normalization by the total power disproportionately affect low-frequency power in the FXS group? We agree with the reviewer that the use of relative power, which is commonly reported in human studies to normalize between subjects, needs further justification in clinical populations.

We have addressed this concern in detail in the results (“Diminished alpha power in FXS is not dependent on elevated gamma power”) supported with new analyses and depicted in updated figures. The conclusion from these results is that 1) alpha power reduction in FXS is not dependent on gamma power (either at the electrode or source level) and 2) a systematic shift in absolute power in participants with FXS supports the use of relative power normalization to address baseline bias.

Electrode-level analysis: Spectrogram: Scalp-averaged absolute power spectrograms displayed a marked increase in FXS participants across most frequencies (**see Figure 2C**). A clear decrease in absolute alpha2 power is visible in males with FXS even as each band is treated independently. Topography: We then retained the channel dimension and compared scalp topography (**see Figure S1**). Here, we also observed elevated absolute power across most frequencies. In the alpha range, FXS and controls were generally statistically similar but there were patches of increased alpha1 and decreased alpha2 in FXS.

Source-level: We compared source estimated absolute power between FXS (**see Figure S2**) and controls and a male-only subgroup (**see Figure 2F**). We identified punctate areas in the alpha2 band in which absolute power was significantly decreased at the group level (blue) and more prominent reduction in alpha2 in male-only subgroup. Otherwise, absolute power of the cortical envelope was significantly increased (red) or equivalent (grey) in FXS compared to controls across all frequency bands.

Summary: These comparisons show a large baseline difference in absolute power, such that FXS is larger than controls for most frequencies. Despite this, participants with FXS (exemplified in males) show similar or lower levels of alpha2 power irrespective of activity in other frequency bands. These results support the use of relative power normalization of raw signal to mitigate subject- and group-specific biases. Absolute power is more influenced by factors such as skull thickness and head geometry which vary considerably across participants[1] and such factors are present in FXS[2, 3].

Reviewer #3: Results

Overall, I found the results section to be clear at the beginning, but still dense to read, and progressively less clear (especially starting at the Cognitive Networks section). I recommend presenting the reasoning behind each analysis – what question are the authors hoping to address with the following analysis – to help better guide the reader through the results section. I recognize that many cognitive neuroscience papers simply state the results without the background reasoning provided. Given a large amount of data and analyses presented, the current version lacks clarity in the reasoning

Manuscript Title: Neocortical Localization and Thalamocortical Modulation of Neuronal Hyperexcitability in Fragile X Syndrome

behind the analyses and thus the digestibility of the findings is diminished. I also wonder whether the Cognitive Networks section could be presented in a table format so that the text could focus more on what analysis was performed and why, and then highlights?

Authors' Response A5:

We appreciate the reviewer's forthright feedback regarding the organization of the results section. We agree with the author, given the breadth of the results, we have thoroughly revised the Results section to provide key context to each analysis. As suggested, we made more extensive use of tables and visualizations to present quantitative results and prioritize the text for summary and highlights of the analysis.

Result revision list:

Manuscript

1. Major text revision of the Introduction, Results, and Discussion
2. Figure 1: Clarification regarding vertex, node, region, and RSN level analysis (revised)
3. Figure 2: Absolute power spectrogram (new), F. male subgroup source comparison (new)
4. Figure 3: Regional and RSN power comparison (revised)
5. Figure 4: PAF panel results (new)
6. Figure 5: CFC panel results (revised)
7. Table 2: Sex and Group differences by cortical region (revised)

Supplement

1. Figure S1: Scalp absolute power topography comparison (revised)
2. Figure S2: Source absolute power comparison (new)
3. Table S2: Preprocessing characteristics (new)
4. Table S3: Summary of Source-estimated Peak Alpha Frequency (PAF) by Cortical Node (new)
5. Table S4: Pairwise group differences of gamma1 cross-frequency coupling by cortical node (revision)
6. Table S5: Pairwise group differences of gamma1 cross-frequency coupling by RSN (revision)
7. Table S6-S11: Clinical correlations (revised)
8. Appendix 1: visual depiction of atlas nodes (new)
9. Appendix 2: atlas key for node, region, RSN (revised)

Reviewer #3: Figure 1C

1. Figure 1C: Perhaps this should be plotted differently? The inset graph shows clear increased power between 30-90hz for the FXS group; however, this cannot be seen at all on the right side of the larger plot, presumably because of the y-axis scale that is used. Is there a reason this wasn't all plotted on a log scale?

Authors' Response A6:

Manuscript Title: Neocortical Localization and Thalamocortical Modulation of Neuronal Hyperexcitability in Fragile X Syndrome

We have enhanced the visualization of the spectrogram plots (for relative and absolute power) by using a 1/F normalization (multiply power by frequency) for visualization of gamma activity (**see Figure 2B-C**).

Reviewer #3: Absolute Power

I would like to see the absolute power plots as well. As discussed above, the increase in high-frequency band power could lead to reduced relative power in lower frequencies, while absolute power is similar.

Authors' Response A7:

2. We have addressed these concerns with revisions to the text of the results (see “Diminished alpha power in FXS is not dependent on elevated gamma power” and “Global decrease in alpha activity and elevated gamma activity in temporal lobes”). We have added the following figures specifically to address absolute power:

Figure 1B: Absolute power spectrogram

Figure 1F: Male FXS – Male Control group comparison of source power

Figure S1: FXS-Control scalp topography absolute power contrasts

Figure S2: FXS-Control source absolute power contrasts

Reviewer #3: Relative Power Methodology

3. Figure 1E: I apologize if I missed this in the methods section – is power measured over the band the average relative power/Hz within the frequency range, or the sum of the relative power across the defined frequency range? It would be helpful to have a unit of power so it is clear what is being calculated here.

Authors' Response A8:

To implement relative power normalization, we divided power by the sum of the relative power across the defined frequency range (i.e., the sum of relative power is equal to 1). As relative power is a proportion (band power V^2/Hz divided by total power V^2/Hz), it is usually reported without a unit. The units used to represent the minimal norm source activation are picoampere-meters. We have revised the methods to capture units prior to normalization:

Methods

Electrode Power: “Relative power was defined as the band-specific cumulative absolute power (V^2/Hz) divided by the total power (V^2/Hz) across all defined bands and then averaged over available trials.”

Scalp Power: “Welch’s method was used to estimate spectral power from CSD. Frequency band definitions and relative power normalization were identical to methods to calculate electrode power.”

Reviewer #3: Relative Power Methodology

4. Is it standard practice to use relative power for cross-frequency coupling? I am not an expert in CFC, however since you are already comparing power between two frequency bands within an individual, it seems that absolute power would be more accurate in this instance. By definition, relative power in the alpha band is normalized in part by power in the beta and gamma bands – and this may not occur equally between FXS and controls, given the increase in gamma in the FXS group. In addition, given

the shift in peak from 10 to 7 Hz, it would be more informative to compute comodulograms comparing low and high-frequency ranges, so that the association can be more specifically analyzed across the range of frequencies.

Authors' Response A9:

There is considerable variability in the methodology used to conduct EEG connectivity/coherence analyses, including precedents for calculating cross-frequency power coherence using relative power [4, 5]. Our use of the CFC analysis was focused on 1) providing further insight into the main power analysis of the paper, 2) supporting or refuting evidence of TCD in FXS, and 3) clarifying the spatial distribution of the previous electrode-level CFC results in FXS. Thus, we have conducted our major analyses in relative power in order to simplify interpretation and integration of our results across the manuscript. After reviewing scalp and source absolute power estimates, we are more confident that the use of relative power normalization is appropriate to mitigate interindividual and group-level variability.

Second, we opted to use a “low-resolution” version of CFC to assess whole-brain averaged alpha or theta power to high-frequency power across individual cortical nodes[6]. Though this method is not suitable for a more detailed analysis of point-to-point connectivity or frequency comodulograms, it is well-suited to answer hypotheses derived from our main analyses and compare to reports of TCD in other disorders. Within the scope of TCD, phase amplitude (or phase-phase) coupling is not well-understood. We anticipate a future manuscript with a focus on novel high-resolution spatial and frequency connectivity to explore in FXS, but outside the scientific focus of the current manuscript.

Reviewer #3: Figure 2C

5. Figure 2C – I was surprised that this analysis was done using a ratio of relative power approach. If the question is where topologically the shift to a lower alpha frequency occurs, why not look at the peak frequency across different regional nodes? Similar to my concern above – I'm not convinced that relative power is the best measure for this analysis, since a ratio between alpha power is already naturally normalized to itself.

Authors' Response A10:

We agree with the author that peak alpha frequency (PAF) is a suitable alternative to the alpha ratio approach in the original manuscript. We initially felt an alpha ratio approach may perform better than peak alpha frequency detection across a greater number of subjects and channels/sources. We have addressed any shortcomings with revised code and have presented a revised analysis with PAF. Though the results did not change substantially, we agree with the reviewers that the implementation is more intuitive and easier to interpret. Please see revised results and in **Figure 4**.

Reviewer #3:

6. Oscillatory activity in cognitive and affective network section: “Between and within group comparisons.....” sentence is not complete.

Authors' Response A11:

We have substantially revised this section and thoroughly copy-edited the final copy.

Reviewer #3: Sex differences / mosaicism

7. Can the effects of sex be separated from the effects of mosaicism across both sexes? Given the large sample size, can you do this comparison using only females and mosaic males? I think it is important to know whether this is truly a sex difference versus a difference in protein expression. In

Figure 3b, I would be interested in seeing the difference between mosaic males and full mutation (“obligate mosaic”) females as they are more likely to have similar protein expression deficits.

Authors’ Response A12:

We share the reviewer’s interest and enthusiasm regarding this interesting comparison between females affected by full mutation FXS and males with mosaicism. Though being sufficiently powered to study sex differences (> 25 per group), males with mosaicism are a relatively small subset (n=12). Importantly, mosaic males with FXS can also vary phenotypically [7] based on either repeat number mosaicism or methylation mosaicism which further highlights the importance of a well-powered sample.

Our expectation is to examine this question after additional data collection rather than to present under powered results. To shed some light into this question indirectly, we have performed exploratory clinical correlations with full mutation, non-mosaic male participants.

Reviewer #3: Shift in low-frequency peaks

8. I found the shift in low-frequency peaks from 10 to 7 Hz to be very intriguing, especially in the context of TCD. I would be interested in further discussion on the authors’ thoughts on whether this is a slowed alpha or a shift from the alpha to theta oscillations, and how conceptually these would relate to the TCD construct. On the individual level, is there a single peak, or do individuals have both theta and alpha peaks? I’d also be interested in a discussion of why amplitude-amplitude coupling rather than phase-amplitude coupling was analyzed. Phase-amplitude coupling between theta and gamma is well studied and as I understand, is also associated with TCD. What information do each of these methods provide, and why chose one over the other? It seems that with the differences in high-frequency power in the FXS group, use a phase approach may be helpful.

Authors’ Response A13:

We share the reviewer’s interest in a mechanistic understanding of PAF alterations and have substantially revised the flow of the results and discussion focus. Quantifying individual-level, trial-by-trial variability (non-averaged data) is an emerging and exciting area of EEG research[8], but was outside the scope of our current power-based scientific hypotheses. We are currently addressing several issues (i.e., accounting for epoch noise, missed peak detection) to allow for non-averaged data in future manuscripts with a separate set of hypotheses.

Though power-power coupling was applied in the original TCD model[9], a recent paper has introduced the use of phase-amplitude coupling as a potential more accurate reflection of physiological communication in the brain[10]. Though intriguing, we opted to present power-power coupling, rather than PAC, based on 1) replication and comparison to the original TCD model, 2) adding further context to the power results which are the main focus of the paper, and 3) recent heightened criticism of univariate PAC methods of scalp EEG data which have led to systematic, spurious findings [11-13]. We are investigating several newer methods to address these methodological limitations including multivariate PAC [14] and other methods which may reduce spurious findings from epiphonema found in EEG [15]. Nevertheless, the inclusion of these methods would be primarily exploratory and outside the scope of previous reports. Given these concerns, we opted to focus our presentation of TCD features in FXS primarily centered from a power analysis.

Reviewer #3: Discussion of Sex Differences Observed

9. I would like to see a more thorough discussion of the sex differences observed, especially in the context of mosaicism in both male and females in the dataset. Authors discuss the effects of residual FMRP levels in females and how this may moderate cortical hyperexcitability, however wouldn’t this also be true for mosaic males?

Authors’ Response A14:

As we review in authors' response in A12 and A14, we agree with the reviewer's suggestion and more explicitly stated our tested hypotheses in the results and incorporated additional discussion.

Reviewer #3: Gamma power and cognitive scores

10. What hypotheses do authors have on why associations between gamma and cognitive scores were only present for full mutation males? Wouldn't one expect an even clearer dose response between gamma activity and cognition in the mosaic population if gamma were directly related to cognitive development? Or does this suggest that having no FMRP production leads to different downstream brain differences than having low FMRP production?

Authors' Response A15:

The reviewer's query regarding clinical correlations prompted us to substantially revise our organization and presentation of these results. This includes:

1. Introduction: Prediction of clinical associations with the TCD framework
2. Results: Revision of text now organized by clinical measure. We have presented significant correlations at the regional (anatomical organization) or network (functional organization) within across all FXS subjects and in a subgroup of full-mutation FXS, non-mosaic male participants.
3. Results: Interactive web-based correlation results to examine scatter plots and uncorrected clinical associations (hosted at: https://epedapati.shinyapps.io/tcdfxs_corr/).
4. Discussion: updated interpretation of correlation results and updated limitations regarding mosaic status.

In response to the reviewer's specific requests:

The present correlation results do support a broader FXS relationship of elevated gamma activity with worse cognitive scores and survives 5% FDR correction (**see Table 1 below; scatter plots available on weblink in A15**).

	Measure	Frequency	Location	n	r	p	adj.p
30	NVIQ	gamma1	LC	64	-.39	0.001	0.028
31	NVIQ	gamma1	RO	64	-.39	0.001	0.028
33	NVIQ	gamma1	RT	64	-.39	0.002	0.029
35	NVIQ	gamma2	RT	64	-.39	0.002	0.029
36	NVIQ	gamma2	LC	64	-.38	0.002	0.03
43	NVIQ	gamma1	LT	64	-.37	0.003	0.037
54	NVIQ	gamma2	RC	64	-.35	0.004	0.049

Table 1: Association of non-verbal IQ with regional gamma power in participants with FXS (n=64)

However, the reviewer's observation why this relationship isn't "clearer" than with full-mutation, non-mosaic males, raises interesting speculation. First, the spectral power results demonstrate on average, females with FXS demonstrate similar abnormalities as males in low-frequency bands, but do not have heightened gamma power (**see Figure 3**). Since we use sex-matched controls, this suggests variability in FMRP expression may ultimately lead to cortical changes that also result in variability in gamma power. We speculate that gamma changes seen in full mutation, non-mosaic males may be more directly associated with impairments in cognitive function. Indeed, females with FXS display greater variability in their cognitive profiles and may suggest functional consequences in the cortical distribution and expression of FMRP [16]. As reviewed under response A12, though parsing clinical correlations in

Manuscript Title: Neocortical Localization and Thalamocortical Modulation of Neuronal Hyperexcitability in Fragile X Syndrome

male mosaics (methylation-type and repeat count-type) may more directly answer this question, the present sample is underpowered to support such an analysis. We have incorporated these considerations into the discussion.

Reviewer #3 Minor Concern

11. Page 2 “key illness feature”, I found “illness” to be unusual descriptor as it generally is associated with disease rather than disorder. I might use a different word here – a key characteristic of the disorder? Or maybe hyperexcitability is central to pathophysiology of the disorder?

Authors’ Response A16:

We agree with the reviewer the word disorder is appropriate in this context.

Reviewer #4 (Remarks to the Author): Summary

The paper submitted by Pedapati et al. presents an EEG study in Fragile X syndrome patients, in which the authors want to examine the neocortical localization of thalamocortical modulation of neuronal hyperexcitability. A strength of the study is the large sample size in this rare disease. However, although I think that the authors apply sophisticated EEG methods, I do not see much justification for their analysis, motivation for the study, and conclusions are drawn. Below I outline the most problematic aspects.

Authors’ Response A17

We appreciate the reviewer’s appraisal of our submitted manuscript, and we would appreciate the reviewer’s reconsideration of our revision. We agree on the strengths of the study – the large sample size and intriguing EEG findings are overshadowed by the organization and flow of the writing. Based on reviewer feedback, we have thoroughly revised all aspects of the manuscript to highlight the context and significance of these findings. We have made the following major revisions:

Manuscript

1. Major text revision of the Introduction, Results, and Discussion
2. Figure 1: Clarification regarding vertex, node, region, and RSN level analysis (revised)
3. Figure 2: Absolute power spectrogram (new), F. male subgroup source comparison (new)
4. Figure 3: Regional and RSN power comparison (revised)
5. Figure 4: PAF panel results (new)
6. Figure 5: CFC panel results (revised)
7. Table 2: Sex and Group differences by cortical region (revised)

Supplement

1. Figure S1: Scalp absolute power topography comparison (revised)
2. Figure S2: Source absolute power comparison (new)
3. Table S2: Preprocessing characteristics (new)
4. Table S3: Summary of Source-estimated Peak Alpha Frequency (PAF) by Cortical Node (new)
5. Table S4: Pairwise group differences of gamma1 cross-frequency coupling by cortical node (revision)

6. Table S5: Pairwise group differences of gamma1 cross-frequency coupling by RSN (revision)
7. Table S6-S11: Clinical correlations (revised)
8. Appendix 1: visual depiction of atlas nodes (new)
9. Appendix 2: atlas key for node, region, RSN (revised)

Reviewer #4: Introduction

1. To me in the introduction, the authors mainly motivate their study on the grounds of insufficiencies of previous studies (sample size, un-balanced gender, lack of network modelling). Especially regarding the latter is not clear why(!!) the network modelling is important for an advance in the conceptual understanding of FXS.

Authors' Response A18:

The reviewer raises the insufficiency of the introduction in poorly conveying our rationale behind the premise of the study. We revised the text of the entire introduction to communicate the aims and organization of the study analyses. In terms of network modeling, the cortical parcellations that result from source estimation can be grouped either at the anatomical level or through non-contiguous functional groupings (i.e., resting-state networks). As FXS has some somatic manifestations (and minor structural imaging findings), the hallmark features of the syndrome are primarily developmental and neuropsychiatric. We hypothesized that grouping by resting-state networks, rather than continuous anatomical regions, may better model clinical measures. In addition, network modeling can provide a systems-level perspective of the human cortex. We found evidence of such a discrepancy in our clinical correlations where network-based feature estimates were more closely associated with clinical features than continuous anatomical feature estimates.

Reviewer #4:

2. Related to the above point, the authors strongly draw on the role of gamma oscillations in neurodevelopmental disorders. They seem to equal gamma oscillations with ability in higher-order cognitive functions, which is not true. Lower-frequency oscillations (especially theta oscillation) may be of similar importance, especially given their relevance in integrating processes between distant brain regions (as an aspect relevant to FXS). The authors do not provide justification for their analysis of various frequency bands and the entire data analysis is very exploratory. Similarly, there is no conceptual justification for e.g., analysis of alpha-gamma coupling vs. theta-gamma coupling. To go on; what is the conceptual relevance for the various correlation analyses performed between EEG frequency band power and clinical parameters such as OCD symptoms, irritability, etc.?

Authors' Response A19:

The reviewer raises concerns that the analytic plan was primarily exploratory, with little justification for the analysis of frequency bands. We have revised the text of the introduction, results, and discussion to address this concern.

The selection of frequency bands, with an emphasis on theta, alpha, and gamma activity was based on *a priori* hypotheses from previous human literature and translational studies. The earliest clinical EEG studies in FXS found global theta and alpha changes which were replicated by subsequent studies[17] and our premise centered around the thalamocortical system. The more recent finding of elevated non-myogenic gamma activity was particularly interesting as it is parallel to similar findings in the *Fmr1^{-/-}* KO[18]. As the novel treatment develop in FXS is centered around the *Fmr1^{-/-}* KO, there remains a significant gap in parallel electrophysiology between the animal model and human studies. However,

we share the reviewer's view that gamma power cannot be viewed as a marker of higher-order cognition. The role of gamma activity is increasingly complex, and this has been revised in the introduction and discussion. Thus, rather than an exploratory endeavor, we sought specifically to 1) confirm or refute evidence to support a systems-level TCD model within FXS and 2) describe the distribution of elevated gamma oscillations (anchored in parallel findings in the *Fmr1*^{-/-} KO[18, 19]).

Reviewer #4:

3. Generally, it is not clear why the different analyses presented in the results section were performed and how these conceptually relate to each other and the main study question at hand. I think that for clinical studies of this sort there should be clear conceptual hypotheses that should be focused. Especially considering 'replication crisis' a data-driven convenience approach as performed by the authors bears the risk that non-meaningful results become published. Generally, the results section is very difficult (if not impossible to understand), because the different steps undertaken in the data analysis are not justified/unclear. Just one example: "Prominent group differences (FXS-Control) in RSN activity by frequency band emerged at the node level in DMN". Which other networks? Why to the authors focus on the "node level"? Why is a frequency band specific analysis reasonable/justified? Why do the authors stress the DMN? What about the other networks/why are these less interesting to report in more detail? Similar lists of opaque analysis steps can be given for almost all sections of the results.

Authors' Response A20:

Based on reviewer feedback, we have revised the analysis plan in the Results section and listed specific hypotheses being tested and why these queries were generated. We have extensively revised Figures and Tables as outlined in authors' response A17 to address the observed concerns directly. We are hopeful these extensive edits and additional analyses, including justifications for each analysis, address these overall concerns.

Reviewer #4: Concern about ability to replicate our work

4. Frank observations regarding the results section and the larger concern that hypothesis free science can lead to results that can not be replicated.

Authors' Response A21:

We did not take these concerns lightly and have thoroughly restructured the results section to add context, in addition to new analyses. The revised section is more in line with the progression of how we tested each hypothesis.

Reviewer #4: Too much speculation in the discussion section

5. Large parts of the discussion section are too much speculative. For example, the authors extensively discuss the role of GABAergic cells, but the data level analyzed allows little insights into this level of arguments. The authors do not convey a clear picture on how the different analyses performed convey to a better understanding of FXS. This is, I think, related to the exploratory nature of the study.

Authors' response A22:

We have restructured and pruned the discussion to more closely align with the results and key hypotheses. We have also removed portions which lack concrete anchors in the text and considered speculation.

Reviewer #4 The model of thalamocortical dysrhythmia proposed is not sufficiently related to the data obtained

6. In relation to the above comment, the hypothetical model of thalamocortical dysrhythmia in Figure 4 is not sufficiently related to the data obtained. In fact, the authors provide data how different EEG spectra are modulated in FXS and what cortical structures are related to it. Their model in Figure 4, however, is cortical layer specific (not measured in the study) and draws on the role of thalamic connections to the cortical layers (also not measured in the study). Therefore, the theoretical impact the authors try to convey in their study is heavily overdrawn. Also, the model is too simplified as it equals the output of layer 5 (not really defined what output it is) to intellectual disability, anxiety etc. This is too simplistic and, I think, misleading.

Authors' response A23:

We appreciate the reviewer's focused feedback regarding the relationship between our data and our main hypothesis regarding TCD. We have substantially restructured our revision to closely align our hypothesis testing with aspects of the TCD model. These revisions include:

1. Having a more central focus of the TCD model and specific predictions in FXS in the introduction.
2. Examining global alpha changes in absolute and relative power at the electrode and source level (**see Figure 2 and S1**)
3. Provided key evidence for three hallmark features of TCD including alterations in low frequency activity and elevated gamma power (see Figure 2), leftward shift of peak frequency (see Figure 4), and predominance of theta-gamma CFC over alpha-gamma CFC (see Figure 5).
4. We have removed areas of extensive speculation (including Figure 4) and more closely aligned our text revision based on these findings.

Reviewer #4 Conclusion:

Overall, I think that the paper needs major reworking in terms of hypotheses and concepts being tested and conclusions drawn from the data analysis. I do not think that the work is publishable as it stands now.

Authors' Response A24:

We appreciate the reviewers time and feedback. We have worked to address these important concerns and appreciate the opportunity to be reconsidered for publication. We believe that our extensive reorganization and other edits to the text combined with the added and refined analyses each which relates to hypotheses tested and described in an organized manner in the text will help to address this important overall reviewer concern.

1. Hagemann, D., et al., *Skull thickness and magnitude of EEG alpha activity*. Clinical Neurophysiology, 2008. **119**(6): p. 1271-1280.
2. Chiu, S., et al., *Early acceleration of head circumference in children with fragile x syndrome and autism*. J Dev Behav Pediatr, 2007. **28**(1): p. 31-5.
3. Turk, J. and M. Patton, *Sensory impairment and head circumference in Fragile X syndrome, Down syndrome and Idiopathic intellectual disability*. Journal of Intellectual & Developmental Disability, 2000. **25**(1): p. 59-68.
4. Bian, Z., et al., *Relative power and coherence of EEG series are related to amnesic mild cognitive impairment in diabetes*. Frontiers in Aging Neuroscience, 2014. **0**.

Manuscript Title: Neocortical Localization and Thalamocortical Modulation of Neuronal Hyperexcitability in Fragile X Syndrome

5. Wang, R., et al., *Power spectral density and coherence analysis of Alzheimer's EEG*. Cognitive neurodynamics, 2015. **9**(3): p. 291-304.
6. Wang, J., et al., *A resting EEG study of neocortical hyperexcitability and altered functional connectivity in fragile X syndrome*. J Neurodev Disord, 2017. **9**: p. 11.
7. Meng, M.L., et al., *The association between mosaicism type and cognitive and behavioral functioning among males with fragile X syndrome*. American Journal of Medical Genetics Part A, 2021.
8. Jones, S.R., *When brain rhythms aren't 'rhythmic': implication for their mechanisms and meaning*. Curr Opin Neurobiol, 2016. **40**: p. 72-80.
9. Llinas, R.R., et al., *Thalamocortical dysrhythmia: A neurological and neuropsychiatric syndrome characterized by magnetoencephalography*. Proc Natl Acad Sci U S A, 1999. **96**(26): p. 15222-7.
10. Vanneste, S., J.J. Song, and D. De Ridder, *Thalamocortical dysrhythmia detected by machine learning*. Nat Commun, 2018. **9**(1): p. 1103.
11. Seymour, R.A., G. Rippon, and K. Kessler, *The Detection of Phase Amplitude Coupling during Sensory Processing*. Frontiers in Neuroscience, 2017. **11**(487).
12. Giehl, J., N. Noury, and M. Siegel, *Dissociating harmonic and non-harmonic phase-amplitude coupling in the human brain*. NeuroImage, 2021. **227**: p. 117648.
13. Aru, J., et al., *Untangling cross-frequency coupling in neuroscience*. Current opinion in neurobiology, 2015. **31**: p. 51-61.
14. Cohen, M.X., *Multivariate cross-frequency coupling via generalized eigendecomposition*. eLife, 2017. **6**: p. e21792.
15. Jurkiewicz, G.J., M.J. Hunt, and J. Żygierewicz, *Addressing Pitfalls in Phase-Amplitude Coupling Analysis with an Extended Modulation Index Toolbox*. Neuroinformatics, 2021. **19**(2): p. 319-345.
16. Schmitt, L.M., et al., *Executive Function in Fragile X Syndrome: A Systematic Review*. Brain Sci, 2019. **9**(1).
17. Musumeci, S.A., et al., *Fragile-X syndrome: a particular epileptogenic EEG pattern*. Epilepsia, 1988. **29**(1): p. 41-7.
18. Jonak, C.R., et al., *Multielectrode array analysis of EEG biomarkers in a mouse model of Fragile X Syndrome*. Neurobiol Dis, 2020. **138**: p. 104794.
19. Goswami, S., et al., *Local cortical circuit correlates of altered EEG in the mouse model of Fragile X syndrome*. Neurobiol Dis, 2019. **124**: p. 563-572.

Reviewers' comments:

Reviewer #3 (Remarks to the Author):

The authors have made substantial changes to the manuscript including reorganization of introduction and additional analyses specifically related to previous concerns raised. However, as a reviewer I have concerns regarding the manuscript that prevent my recommendation for publication. These fall in 3 main domains:

1. Organization, thrust of the paper. While the introduction is greatly improved, the reasoning for framing the results and analyses around TCD is still lacking and I think confusing for readership. Authors either need to expand the introduction to describe in detail what TCD is defined by and how these findings relate to thalamocortical circuitry, or they should fully move the discussion of TCD to the discussion section (ie not including TCD in either the intro or results). In addition, the results section was also improved, however the flow of figures and tables did not always make sense. Often supplemental material is first referenced over main figures and some figures were too densely packed with data to be easily interpretable due to size. As a reviewer this made it difficult to follow and assess and in turn reduced the impact of the findings.

2. Typos/mis-referenced figures. There were several occasions where I noted portions of figures that were never referenced in the paper text, or figures that appear to be mis-referenced in the text. I have listed those I observed below. The number that I observed makes me concerned that there are more that I may have missed.

3. Relative power – the authors have presented additional analysis related to absolute vs relative power. For reasons outline below, I still have concerns that by using relative power specifically for sex difference the findings may not accurately reflect oscillatory similarities/differences in the lower frequency bands. I am not asking that the authors do additional analyses in absolute power, but I do think that the discussion needs to include limitations in interpretation of using relative power especially in the case where the differences in absolute power between groups do not seem to be just in offset of the power spectra, but also in the $1/f$ slope and where peak frequencies are located.

Results:

1. It would be helpful to have the frequency bands defined in the figure or text so that the power spectra and bar graphs can be compared.

2. Table 1: Sex has female in parentheses, but the provides both M and F numbers.

3. While across sex FXS findings are similar for relative and absolute power, it does appear that elevated delta-theta and gamma power in the male FXS group does affect the relative theta power, as there in Figure 2C there appears to be substantial difference between absolute power of the theta peak between FXS males and females, which is lost when looking at relative power. I wonder if you are losing a significant "dose response" aspect to the peak shift and increased amplitude by focusing on relative power.

4. Reference 20 specifically finds that "there was only mediocre association between EEG alpha power at frontal, temporal, and parietal sites and the thickness of the underlying skull....skull thickness may be neglected as a potent source of error when individual brain activity are indexed by magnitude of EEG alpha activity". In addition, given that this study is in adults when skull thickness is more similar than in developing children, I don't think this is appropriate justification for using relative power.

5. Figure 2D and 2E are referenced for vertex level analyses – but I believe Figure 2D is not related to source localization.

6. What do authors make of the increased increases in low and high frequency absolute power that is different between males and females, also suggestive of a dose dependent genetic effect? Is there a reason to think these are not real affects? The relative power source localization findings presented support no difference in low-frequency, but it would seem that there would be differences in absolute power. Outside of head circumference (which per reference 20 does not seem to influence alpha power substantially) is there another reason to suggest that this sex

difference is not important? Much of the discussion centers around sex findings that are specific to gamma and not theta, however this assumes that the differences in absolute power are not biologically relevant. This should be further discussed in the discussion as a limitation of the interpretation of the findings.

7. Figure 4C and associated Node Level text was very confusing. The text says that significant differences were in the left parietal region, but in the figure the darkest areas are shown in the right hemisphere. Table S3 does not list the parietal regions in the text, but instead the right posterior cingulate, and right superior parietal nodes have FDR <0.001.

8. Figure 5C is barely readable. I'm not sure I would keep this in the main part of the manuscript.

9. I don't think Figure 4D is not referenced at all in the text.

Minor Comments

For future submission, it would be very helpful to have line numbers and page numbers in documents to aid the reviewer.

1. The description of near-universal intellectual disability is more accurately a description of males with full mutation and methylation of the Fmr1 gene, which is not the same as having a diagnosis of FXS, as many females with various penetrations also have FXS but do not have intellectual disability.

2. End of paragraph 1 has a comma instead of a period.

3. The first section is titled 'Participants' but it seems to cover more about experimental and statistical design. Would recommend change the title of this section.

4. Figure 2A is not referenced in the text. It certainly is important to the description of the participants, so perhaps it should be directly referenced?

5. It seems strange to lead each power section with findings that are then shown in the supplemental materials rather than the main manuscript, and this makes it harder for the reader to follow/interpret results. Recommend starting with findings that are shown in the manuscript.

6. I believe that findings shown in Figure 2D are referenced in the text but the figure itself is not referenced.

7. Description of source localization starting with "We also employed...." is not clearly described.

Reviewer #4 (Remarks to the Author):

I thank the authors for being responsive to my comments. I think that the current version of the manuscript has been much improved the work has gained clarity. I think it is publishable in Comms Biol.

Neocortical Localization and Thalamocortical Modulation of Neuronal Hyperexcitability in Fragile X Syndrome

Response to Reviewers

March 16, 2022

Reviewer #3: Summary Feedback

“The authors have made substantial changes to the manuscript including reorganization of introduction and additional analyses specifically related to previous concerns raised. However, as a reviewer I have concerns regarding the manuscript that prevent my recommendation for publication. These fall in 3 main domains:”

Authors’ Response 1: We appreciate the reviewer’s constructive feedback and have worked to thoroughly address the concerns within these three domains. We feel strongly these revisions have greatly strengthened the clarity, thrust, and rigor of the manuscript.

Review #3 Detailed Feedback

Organization, the thrust of the paper: “While the introduction is greatly improved, the reasoning for framing the results and analyses around TCD is still lacking and I think confusing for readership. Authors either need to expand the introduction to describe in detail what TCD is defined by and how these findings relate to thalamocortical circuitry, or they should fully move the discussion of TCD to the discussion section (ie not including TCD in either the intro or results).”

Authors’ Response 2: We agree with the reviewer’s recommendation that a greater emphasis on TCD within the introduction would clarify the thrust of the paper and the rationale behind our analysis approach. To address this:

1. Substantial revision of paragraph 3 of the introduction (and edits within the discussion) to clarify the 1) electrophysiological definition of TCD, 2) supportive details regarding thalamocortical circuitry, and 3) previous reports of abnormal thalamic activity in FXS which led to our hypothesis.
 2. Expanded discussion of TCD within the discussion (page 11), added further physiological and molecular ties to TCD in FXS (page 13)
 3. Revised the conclusion paragraph (page 13).
-

Reviewer 3: “In addition, the results section was also improved, however the flow of figures and tables did not always make sense. Often supplemental material is first referenced over main figures and some figures were too densely packed with data to be easily interpretable due to size. As a reviewer this made it difficult to follow and assess and in turn reduced the impact of the findings.”

Authors’ Response 3: We acknowledge the reviewer’s concerns and have responded individually with detailed feedback in the responses below.

Reviewer 3: "Table 1: Sex has female in parentheses, but the provides both M and F numbers."
Authors' Response 5 Column labels have been corrected.

Reviewer 3: "While across sex FXS findings are similar for relative and absolute power, it does appear that elevated delta-theta and gamma power in the male FXS group does affect the relative theta power, as there in Figure 2C there appears to be substantial difference between absolute power of the theta peak between FXS males and females, which is lost when looking at relative power. I wonder if you are losing a significant "dose-response" aspect to the peak shift and increased amplitude by focusing on relative power."

Authors' Response 6: We agree with the reviewer's thoughtful consideration of the revised findings. This discussion highlights the importance of presenting both absolute and relative power in parallel. In our opinion, the FXS male (M) and female (F) difference in absolute power represents a parallel, inflated version of M-F differences in the control group tracings, akin to the overall baseline shift in absolute power between groups. The largest differences remain in the alpha band where a difference between males and females is expected due to head circumference. Thus, the differences observed in absolute power may be biologically meaningful in that they represent real differences based on sex and/or head circumference, but do not necessarily reflect a unique group effect. The current presentation does, however, allow the reader to draw inferences regarding group/sex differences from absolute and relative power perspectives from the dataset. We have made the following changes to further address the reviewer's concerns (**see Authors' Response 9 for additional related changes**):

1. Substantial revision of discussion (page 12) to expound on implications of low and high-frequency changes between sexes in FXS. We also proposed how similar theta (after correction) in females with FXS, but varying alpha and gamma power be associated with clinical findings.
 2. Updated limitations to address the reviewer's concern regarding absolute and relative power differences.
-

Reviewer 3 "Reference 20 specifically finds that "there was only mediocre association between EEG alpha power at frontal, temporal, and parietal sites and the thickness of the underlying skull....skull thickness may be neglected as a potent source of error when individual brain activity are indexed by magnitude of EEG alpha activity". In addition, given that this study is in adults when skull thickness is more similar than in developing children, I don't think this is appropriate justification for using relative power.

Authors' Response 7: We appreciate the reviewer's feedback and wanted to add additional context. Though skull thickness was one consideration for using relative power, our primary rationale was to minimize systematic differences related to group and age effects (our study encompassed both adults and children) associated with head circumference. In the literature, head circumference is associated with alterations in alpha rhythm including power and peak alpha frequency (Nunez 1978; reference added to the manuscript). We have also revised our limitations to more clearly address this concern.

Reviewer 3 Figure 2D and 2E are referenced for vertex level analyses – but I believe Figure 2D is not related to source localization.

Authors' Response 8: Figure 2D and 2E do represent vertex-level, source-localized analyses. Figure 2D and 2E have been swapped to clarify the derivation of boxplot values from the significant contrast regions of the vertex-level comparison. The caption to Figure 2 has also been updated.

Reviewer 3: What do authors make of the increased increases in low and high frequency absolute power that is different between males and females, also suggestive of a dose dependent genetic effect? Is there a reason to think these are not real affects? The relative power source localization findings presented support no difference in low frequency, but it would seem that there would be differences in absolute power. Outside of head circumference (which per reference 20 does not seem to influence alpha power substantially) is there another reason to suggest that this sex difference is not important? Much of the discussion centers around sex findings that are specific to gamma and not theta, however, this assumes that the differences in absolute power are not biologically relevant. This should be further discussed in the discussion as a limitation of the interpretation of the findings.

Authors' Response 9: Yes, the authors agree with the reviewer's interpretation that low and high-frequency within-sex alterations in FXS represent biologically relevant findings and serve as a proxy of Fmr1 genetic liability. We believe our original manuscript over-emphasized gamma sex differences but did not sufficiently address findings in the theta/alpha band.

To review, we first looked at group-only effects by high-resolution vertex comparison of source models (**see Figure 2**), but then follow up this significant finding with additional models examining group X sex effects in the region and network models (**see Figure 3C and Results "Modulation of gamma power abnormalities by sex"**).

The reviewer specifically identified a discrepancy in FXS sex differences in the theta band between relative and absolute power (**see Figure 2B and 2C**). We, as the reviewer, also predict theta power based on genetic liability would be higher in males than females with FXS. Indeed, even in our relative power contrasts (**see Figure 3C**) the same directional effect is visually present [as in absolute power] but did not reach statistical significance after multiple comparisons. Additionally, relative and absolute spectrograms demonstrate within band variability between the theta band (3.5 to 7.5 Hz). The absolute power spectrogram (**see Figure 2C**) demonstrates within band variability, but also a large magnitude difference between sexes. Conversely, the relative power spectrogram (**see Figure 2C**) maintains the within band variability (approximately 6 Hz), but the magnitude difference is diminished. We speculate if the theta band was split into low and high frequencies (which is not commonly done in human EEG studies) relative power sex effects in theta band may be detected. The present study does, however, present the reader with a comprehensive view through either absolute or relative power perspectives of theta band findings in FXS. We have also added to the discussion the potential implications of males and females with FXS having similar theta power (shared liability for dysregulation of thalamocortical signaling), but females with FXS have increased alpha power (protective effect of expression of FMRP). This may explain the differential findings in gamma power and also be related to the variation in clinical correlations between gamma and clinical findings.

To clarify our findings to avoid confusion to readers, we have made several revisions to the manuscript:

1. Abstract: Added low and high-frequency sex effects to the abstract
 2. Results: Clarified purpose of vertex level testing was to identify group effects with follow-up testing of the contribution of sex by region and network models.
 3. Clarified Results subheading to including sex differences in alpha power
 4. Revised discussion to explicitly mention alpha and gamma sex differences were present. We also substantially revised aspects of expounding and clarifying sex differences (**see Response 6**)
 5. Added points from this response to the limitations (Discussion) section.
-

Reviewer 3: “Figure 4C and associated Node Level text was very confusing. The text says that significant differences were in the left parietal region, but in the figure the darkest areas are shown in the right hemisphere. Table S3 does not list the parietal regions in the text, but instead the right posterior cingulate, and right superior parietal nodes have $FDR < 0.001$.”

Authors’ Response 10: The reviewer noted clarity issues with Figure 4C and potential discrepancies with the associated table, including omission of parietal regions in the Table.

Based on the reviewer feedback, we identified an error in which right (R) and left (L) labels were used inconsistently between panels 4C and 4D. We reorganized the figures to show the same brain orientation in both panels 4C and 4D and added individual R and L labels for each.

The correction to the R and L side figures also resolves the discrepancy with the table. “Parietal” is not listed in the table, but rather “Superior Parietal” and “Inferior Parietal” per the DK atlas convention. The significant regions now correspond to the figure. We appreciate the Reviewer’s careful attention to this detail.

Reviewer 3: Figure 5C is barely readable. I’m not sure I would keep this in the main part of the manuscript.

Authors’ Response 11: We agree with the reviewer that the Figure 5 panel suffers from readability issues. To address this, we have reorganized and rescaled Figure 5 into the same dimensions as Figure 1. We feel this has greatly enhanced the readability of the subfigures.

Reviewer 3: “I don’t think Figure 4D is not referenced at all in the text.”

Authors’ Response 12: We have updated the Results (Clinical Correlations) to reference Figure 4D.

Reviewer 3: Minor Comments For future submission, it would be very helpful to have line numbers and page numbers in documents to aid the reviewer.

Authors’ Response 13: We have updated the document to have page and line numbers to improve readability.

Reviewer 3: "The description of near-universal intellectual disability is more accurately a description of males with full mutation and methylation of the Fmr1 gene, which is not the same as having a diagnosis of FXS, as many females with various penetrance also have FXS but do not have intellectual disability."

Authors' Response 14 We appreciate the reviewer's recommendation in regard to the introduction and have modified the text.

Reviewer 3: "End of paragraph 1 has a comma instead of a period."

Authors' Response 15: Corrected and the paper has been through an additional copy edit.

Reviewer 3: "The first section is titled 'Participants' but it seems to cover more about experimental and statistical design. Would recommend change the title of this section."

Authors' Response 16: We agree and changed the title of the section.

Reviewer 3: "Figure 2A is not referenced in the text. It certainly is important to the description of the participants, so perhaps it should be directly referenced?"

Authors' Response 17: We have corrected this oversight and referenced Figure 2A when describing participants.

Reviewer 3: "It seems strange to lead each power section with findings that are then shown in the supplemental materials rather than the main manuscript, and this makes it harder for the reader to follow/interpret results. Recommend starting with findings that are shown in the manuscript."

Authors' Response 18: We agree with the reviewer and have modified power section in the results to start with the manuscript figures followed by supplemental figures.

Reviewer 3: "I believe that findings shown in Figure 2D are referenced in the text but the figure itself is not referenced."

Authors' Response 19: A reference to Figure 2D is present in the vertex-level Results (page 7, line 193).

Reviewer 3: "Description of source localization starting with "We also employed..." is not clearly described."

Authors' Response 20: We have revised the text on page 6/7 to articulate the atlas grouping strategy more clearly.

Reviewer 3: “Relative power – the authors have presented additional analysis related to absolute vs relative power. For reasons outlined below, I still have concerns that by using relative power specifically for sex difference the findings may not accurately reflect oscillatory similarities/differences in the lower frequency bands. I am not asking that the authors do additional analyses in absolute power, but I do think that the discussion needs to include limitations in interpretation of using relative power, especially in the case where the differences in absolute power between groups do not seem to be just in offset of the power spectra, but also in the $1/f$ slope and where peak frequencies are located.”

Authors' Response: We appreciate the reviewer's detailed concerns regarding the use of relative power. We have addressed some of these concerns directly in **Authors' responses 6, 7, and 9**. We have also additionally updated the limitations to provide additional perspective to the readers.

REVIEWERS' COMMENTS:

Reviewer #3 (Remarks to the Author):

I thank the authors for their detailed response. All of my concerns have been addressed by the changes made to the manuscript.

I did note a typo in the first sentence of the Limitations section - I believe it should say "In general, absolute and relative low and high-frequency power changes..."

As well as a typo in the Conclusions - The second sentence needs to be capitalized.

I look forward to this paper being published! It is a great contribution to the field of FXS research and neurodevelopmental disorders in general.